# Technical Note: Preventing $CO_2$ overestimation from mercuric or copper (II) chloride preservation of dissolved greenhouse gases in freshwater samples

François Clayer[1*], Jan Erik Thrane[1], Kuria Ndungu[1], Andrew King[1], Peter Dörsch[2], Thomas Rohrlack[2]

[1]Norwegian Institute for Water Research (NIVA), Økernveien 94, 0579 Oslo, Norway

[2]Faculty of Environmental Sciences and Natural Resource Management, Norwegian University of Life Sciences, PO Box 5003, 1432 Ås, Norway

[*]Corresponding author(s): François Clayer (francois.clayer@niva.no)

**Abstract**

The determination of dissolved gases ($O_2$, $CO_2$, $CH_4$, $N_2O$, $N_2$) in surface waters allows to estimate biological processes and greenhouse gas fluxes in aquatic ecosystems. Mercuric chloride ($HgCl_2$) has been widely used to preserve water samples prior to gas analysis. However, alternates are needed because of the environmental impacts and prohibition of mercury. $HgCl_2$ is a weak acid and interferes with dissolved organic carbon (DOC). Hence, we tested the effect of $HgCl_2$ and two substitutes (copper (II) chloride – $CuCl_2$ and silver nitrate – $AgNO_3$), as well as storage time (24h to 3 months) on the determination of dissolved gases in low ionic strength and high DOC water from a typical boreal lake. Furthermore, we investigated and predicted the effect of $HgCl_2$ on $CO_2$ concentrations in periodic samples from another lake experiencing pH variations (5.4–7.3) related to *in situ* photosynthesis. Samples fixed with inhibitors generally showed negligible $O_2$ consumption. However, effective preservation of dissolved $CO_2$, $CH_4$ and $N_2O$ for up to three months prior to dissolved gas analysis, was only achieved with $AgNO_3$. In contrast, $HgCl_2$ and $CuCl_2$ caused an initial increase in $CO_2$ and $N_2O$ from 24h to 3 weeks followed by a decrease from 3 weeks to 3 months. The $CO_2$ overestimation, caused by $HgCl_2$-acidification and shift in the carbonate equilibrium, can be calculated from predictions of chemical speciation. Errors due to $CO_2$ overestimation in $HgCl_2$-preserved water, sampled from low ionic strength and high DOC freshwater that are common in the northern hemisphere, could lead to an overestimation of the $CO_2$ diffusion efflux by a factor of >20 over a month, or a factor of 2 over the ice-free season. The use of $HgCl_2$ and $CuCl_2$ for freshwater preservation should therefore be discontinued. Further testing of $AgNO_3$ preservation should be performed under a large range of freshwater chemical characteristics.

**Key-words**: lake, greenhouse gases, water sample preservation, mercuric chloride, metal toxicity,
carbon dioxide
**Running tile**: $CO_2$ overestimation from $HgCl_2$ fixation

**1 Introduction**

The determination of dissolved gases by gas chromatography from water samples collected in the
field allows the estimation of biological processes in aquatic ecosystems such as photosynthesis and
oxic respiration (O$_2$, CO$_2$), denitrification (N$_2$, N$_2$O) and methanogenesis (CH$_4$). This technique is also
useful to test the calibration of *in-situ* sensors in long term deployment. However, the accuracy of this
approach largely depends on the effectiveness of sample fixation. In fact, the partial pressure of the
dissolved gases will continue to evolve in the water sample from the time of collection to the time of
analysis unless biological activity is prevented. This is an issue when field sites are far from
laboratory facilities, and when samples need to be stored until the end of the field season for more
efficient processing in large batches. Hence, before using a given chemical to preserve water samples,
it must be ensured that it is efficient in inhibiting biological activity without changing the sample's
chemistry.
Mercuric (II) chloride (HgCl$_2$) has been widely used as an inhibitor of the above-mentioned biological
processes to preserve water samples for the determination of dissolved CO$_2$ in seawaters (e.g.
Dickson, Sabine & Christian, 2007) and several dissolved gases in natural and artificial freshwater
bodies (e.g. O$_2$, CO$_2$, CH$_4$, N$_2$ and/or N$_2$O; Guérin et al., 2006; Hessen et al., 2017; Hilgert et al.,
2019; Okuku et al., 2019; Schubert et al., 2012; Xiao et al., 2014; Yan et al., 2018; Yang et al., 2015)
because it proved effective at very low concentrations compared to other reagents (e.g. Horvatić &
Peršić, 2007; Hassen *et al.*, 1998). Worldwide efforts have sought to reduce the use of mercury
because it is considered toxic to the environment and exposure can severely affect human health
(Chen et al., 2018). Therefore, alternative preservation techniques to HgCl$_2$ treatment have been tested
for dissolved inorganic carbon (DIC) and δ$^{13}$C-DIC such as acidification with phosphoric acid
(Taipale & Sonninen, 2009) or a combination of filtration and exposure to benzalkonium chloride or
sodium chloride (Takahashi *et al.*, 2019). Previous studies showed that simple filtration (and cooling),
fixation (precipitation) or acidification were effective in preserving water samples (Wilson, Munizzi
& Erhardt, 2020). An alternative to using preservatives is to collect in-situ water samples, extract the
headspace in the field, and analyze the headspace in a laboratory (e.g., Cole et al., 1994; Karlsson et
al., 2013; Kling et al., 1991). However, these techniques were not tested for the simultaneous
determination of several dissolved gases, including CH$_4$ which is subject to rapid degassing during
handling or storage if samples are not preserved because of its low solubility in water (Duan & Mao,
2006). In addition, some of the existing alternatives, such as filtration or field headspace equilibration,
are difficult to operate in remote areas in the field under harsh weather conditions and prone to
potential ambient air contamination. Solutions for water sample preservation should therefore involve
a minimum of manipulation steps in the field to avoid gas exchange with ambient air. Preservative
amendments into sealed water bottles appears as one of the most efficient methods. Copper(II)
chloride (CuCl$_2$) and silver nitrate (AgNO$_3$), the most toxic form of silver, are relevant alternatives to
HgCl$_2$ given their known toxicity (e.g., Ratte 2009; Amorim and Scott-Fordsmand 2012) and wide
application in water treatments and water purification (Larrañaga et al., 2016; Nowack et al., 2011;
NPIRS, 2023; Ullmann et al., 1985). Nevertheless, the efficiency of these alternative preservatives has
never been tested for dissolved gas samples preservation.
The addition of HgCl$_2$ to water is known to produce hydrochloric acid through hydrolysis (Ciavatta &
Grimaldi, 1968) and to form complexes with many environmental ligands, both inorganic (Powell *et*
*al.*, 2004) and organic (Tipping, 2007; Foti *et al.*, 2009; Liang *et al.*, 2019; Chen *et al.*, 2017). The
complexation of Hg$^+$ with the carboxyl or thiol groups of DOC in oxic environments could further
increase the concentration of H$^+$ (Khwaja et al., 2006; Skyllberg, 2008). This acidification can be an
issue in poorly buffered water (low ionic strength) with high concentration of DOC where a shift in
the pH and carbonate equilibrium can be induced. In that case, the estimated CO$_2$ concentration would
be higher after HgCl$_2$ fixation than the *in situ* concentration, and if the shift in pH is not accounted for,
can result in an overestimation of dissolved CO$_2$ and bicarbonate concentrations. A similar
acidification effect is also expected with CuCl$_2$ treatments (Rippner et al., 2021), but not for AgNO$_3$
treatments. Such effects would not be expected in marine water due to the high ionic strength of the
water (Chou *et al.*, 2016) or freshwater with low pH (<5.5) under which conditions nearly all
dissolved inorganic carbon is CO$_2$ (Stumm & Morgan, 1981). Thus, there are clear limits of the
application of HgCl$_2$, and possibly CuCl$_2$, for freshwater sample preservation given its risk of leading
to overestimation of CO$_2$ and bicarbonate concentrations, in addition to exposing field workers to the
risks of its high toxicity.

Here we combine data from laboratory experiments (i) and field work (ii) to illustrate risks of mis-
estimation of dissolved gas concentrations in freshwaters with some preservatives and provide
recommendation for best practices in the field. First, we (i) performed some short-term and long-term
incubations of water from a typical heterotrophic unproductive boreal lake with circumneutral pH,
low ionic strength (poor buffering capacity) and high DOC concentration to test the effect of storage
time and different preservative treatments on the determination of five dissolved gases (O$_2$, CO$_2$, CH$_4$,
N$_2$ and N$_2$O) by headspace equilibration and gas chromatography. The preservatives were mercuric
chloride (HgCl$_2$) and two alternative inhibitors, chosen for their wide and effective application in
water treatments and water purification (copper (II) chloride – CuCl$_2$ and silver nitrate – AgNO$_3$; Xu
& Imlay, 2012;  Rai, Gaur & Kumar, 1981). Unamended water samples, where only ultrapure water
was added, were also included for comparison. In addition, we (ii) analysed dissolved CO$_2$
concentration data obtained from a typical productive boreal lake using two independent methods, one
by gas chromatography following HgCl$_2$ fixation, and one through dissolved inorganic carbon
determination without fixation. We show that the overestimation of dissolved CO$_2$ concentrations
caused by HgCl$_2$ fixation can be predicted based on chemical equilibria.

**2. Methods**
The detailed experimental procedures for investigating (i) the effects of storage time and different
inhibitors on dissolved gas concentrations as well as (ii) the effects of HgCl$_2$ on dissolved CO$_2$
analyses over a range of pH values are summarized in Fig. 1 and described below.
2.1. Effects of storage time and inhibitors on the quantification of dissolved gases
*Study site and sampling*
Surface water was collected from Lake Svartkulp (59.9761313 N, 10.7363544 E; Southeast Norway)
north of Oslo, Norway, on the 4[th] of September 2019.  A 5 L plastic bottle was gently pushed into the
water and progressively tilted to let the water flow into the bottles without bubbling. The bottle
aperture was covered with a 90 µm plankton net to avoid sampling large particles. This procedure was
repeated five times to yield a total water volume of 25 L. The 5 L water bottles were immediately
brought back to the lab. Upon arrival at the laboratory, after temperature equilibration, water from the
5 L bottles was slowly poured, to limit gas exchange with the ambient air, into a 25 L tank to provide
a single bulk sample to start the incubation experiment. Filtration, e.g., with 0.45 or 0.2 µm filters,
was avoided to minimize changes in dissolved gas concentrations (e.g., Magen et al., 2014). The
mixed water sample (25 L) was sub-sampled (0.5 L) for the determination of alkalinity (127 µmol L$^{-1}$
), pH (6.73), ammonium (3 µg N L$^{-1}$), nitrate (5 µg N L$^{-1}$), total N (230 µg N L$^{-1}$), phosphate (1 µg P
L$^{-1}$), total P (9 µg P L$^{-1}$) and TOC (8.9 mg C L$^{-1}$) all analysed by standard methods at the accredited
Norwegian Institute for Water Research (NIVA) lab (see Tab. S1). *In situ* temperature of the lake
water was measured with a handheld thermometer and was 18.5 °C. Note that particulate organic
carbon is a negligible fraction of TOC in Norwegian lake waters, representing on average less than
3% (de Wit et al., 2023).
Lake Svartkulp was selected for this experiment because it is representative of low ionic strength
Northern Hemisphere lakes, typically found in granitic bedrock regions in North-East America and
Scandinavia. It is a typical low-productivity, heterotrophic, slightly acidic to neutral, moderately
humic lake. Similar lakes are found in Southern Norway (de Wit et al., 2023), large parts of Sweden
(Valina et al. 2014), Finland, Atlantic Canada (Houle et al., 2022), Ontario, Québec, and North-East
USA (Skjelkvåle and de Wit 2011; Weyhenmeyer et al., 2019).
*Laboratory incubation experiment with different preservatives and storage times*
The experimental design involved to incubate 72 borosilicate glass bottles (120 mL) filled with lake
water from our 25 L bulk sample subjected to four different treatments: addition of 240μL of a
preservative solution of (i) $HgCl_2$, (ii) $CuCl_2$ or a (iii) $AgNO_3$, or addition of 240 μL of (iv) MilliQ
water. The bottles amended with MilliQ water are hereafter referred to as "unfixed". The 72 bottles
were divided into three groups which were incubated cold (+4°C) and dark for 24h, three weeks or
three months respectively, before being processed for dissolved gas analysis by gas chromatography.
These incubation times were selected to represent situations where samples are processed directly
upon return to the laboratory (24h), or after medium (3 weeks) to long (3 months) -term storage,
respectively. At each time point and for each treatment, a group of 6 bottles were further processed for
dissolved gas analysis. Concentrations of $O_2$, $N_2$, $N_2O$, $CO_2$ and $CH_4$ were determined by gas
chromatography (see below) using the headspace technique following Yang *et al.* (2015). pH was not
measured at the end of the storage period.
In details, within 3h of lake water sampling, the 120mL bottles were gently filled with water from the
mixed sample (25 L). Each 120mL bottle was slowly lowered into the water and progressively tilted
to let the water flow into the bottle without bubbling. The bottle was then capped under water with a
gas tight butyl rubber stopper after ensuring that there were no air bubbles in the bottle.  The bottles
were randomized prior to preservative or MilliQ treatment. The preservative or MilliQ amendment
was pushed in each bottle with a syringe and needle through the rubber septum. To avoid
overpressure, another needle was placed through septum at the same time, at least 2 cm above the
other needle, to allow an equivalent volume of clean water to be released.
Stock solutions of $HgCl_2$, $CuCl_2$ and $AgNO_3$ were prepared according to Tab. 1 using high accuracy
chemical equipment (e.g., high accuracy scale, volumetric flasks). The Ag (Silver nitrate EMSURE®
ACS; Merck KGaA, Germany) Cu (Copper(II) chloride dihydrate; Merck Life Science ApS, Norway)
and Hg (Mercury(II) chloride; undetermined) salts were dissolved in MilliQ ultrapure water (>18 MΩ
cm). For measurement of $CO_2$ in seawater samples, the standard method involves poisoning the
samples by adding a saturated $HgCl_2$ solution in a volume equal to 0.05-0.02% of the total volume
(Dickson 2007). We used this as a starting point and added 0.02 % saturated $HgCl_2$ solution to 18
bottles (240 μL of $HgCl_2$ 10× diluted saturated solution), resulting in a sample concentration of 14 μg
$HgCl_2$ $mL^{-1}$ (51.6 μM; Tab. 1). Based on estimated toxicity relative to Hg (Deheyn et al., 2004; Halmi
et al., 2019), the silver and copper salts were added in molar concentrations equal to two and three
times the molar concentration of $HgCl_2$, respectively (Tab. 1), although it varies between species of
microorganisms and environmental matrices (Hassen *et al.*, 1998;  Rai, Gaur & Kumar, 1981).
*Additional 24h incubation experiment with different preservatives for pH measurements*
Since pH was not measured at the end of the first incubation experiment, we performed an additional
experiment to document any potential rapid (within 24h) impacts of preservative on pH. A total of 48
borosilicate glass bottles (120 mL) filled with lake water were subjected to the same four different
treatments as the first experiment described above: $HgCl_2$, $CuCl_2$, $AgNO_3$ or MilliQ water
amendments. To this end, a 20L water tank was filled with surface water from Lake Svartkulp on the
14th of December 2023. The water tank was immediately returned to the laboratory and left for 24h to
equilibrate to the room temperature. On December 15th, 120mL bottles were gently filled with water
from the bulk 20L sample, as described above. The bottles were randomized prior to preservative or
MilliQ treatment performed as described above. The bottles were then incubated at room temperature
for 2h or 24h. pH was measured in the initial unamended lake water, in 24 bottles opened after 2h
incubation, and in 24 bottles opened after 24h incubation. pH measurements were performed with a
WTW Multi 3620 pH meter calibrated using a two-point calibration at pH = 4 and pH = 7. All pH
measures were corrected for temperature. Water temperature of the water samples during pH
measurements ranged between 19.1 and 21.2°C.

2.2. Effects of $HgCl_2$ on dissolved $CO_2$ analyses over a range of pH values
*Study site and sampling*
Water samples were collected from Lake Lundebyvannet located southeast of Oslo (59.54911 N,
11.47843 E, Southeast Norway). Two sets of samples were taken from 1, 1.5, 2 and 2.5 m depth
using a water sampler once or twice a week between April 2020 and January 2021 for the
determination of (i) dissolved $CO_2$ by GC analysis following fixation with $HgCl_2$ and (ii) DIC
analysis with a TOC analyser. Samples for GC analysis were filled into 120 mL glass bottles (as
described above for the 72 incubation bottles), which were sealed with rubber septa under water
without air bubbles. Samples for GC analysis were preserved in the field by adding a half-saturated (at
20°C) solution of $HgCl_2$ (150 µL) through the rubber seal of each bottle using a syringe, as described
above the 72 incubation bottles, resulting in a concentration of 161 µM similar to previous studies
(Clayer et al., 2021; Hessen et al., 2017; Yang et al., 2015). Samples for DIC analysis were filled
without bubbles in 100 ml Winkler glass bottles that were sealed airtight directly after sampling.
These samples were not fixed in any way and were analysed by a TOC analyzer within two hours.
Note that estimation of dissolved $CO_2$ concentrations from pH and DIC is the least uncertain method
of indirect $CO_2$ concentration with estimated relative error of 6% or less (Golub et al., 2017). Lake
water temperature and pH were measured *in-situ* using HOBO pH data loggers placed at 1, 1.5, 2 and
2.5 m (Elit, Gjerdrum, Norway).
Lake Lundebyvannet has a surface area of 0.4 $km^2$ and a maximum depth of 5.5 m. It often
experiences large blooms of G. semen over the summer between May and September (Hagman et al.,
2015; Rohrlack, 2020). The lake water is characterised by high and fluctuating concentrations of
humic substances (with DOC concentrations ranging from 8 to 28 mg C $L^{-1}$), ammonium (5 to 100 µg
N L$^{-1}$), nitrate (20 to 700 µg N L$^{-1}$), total N (average of 612 µg N L$^{-1}$), phosphate (2 to 4 µg P L$^{-1}$),
total P (average of 28 µg P L$^{-1}$; Rohrlack et al., 2020; Hagman et al., 2015), a fluctuating pH (from 5.5
to 7.3), weak ionic strength with alkalinity ranging between 30 and 150 µmol L$^{-1}$, and electric
conductivity varying from 40 to 70 µS cm$^{-1}$. For more details, see Rohrlack *et al.* (2020).
Lake Lundebyvannet was selected for this experiment because it is representative of productive, low-
ionic strength Northern Hemisphere lakes typically found in the southern part of granitic bedrock
regions in North-East America and Scandinavia.
2.3. Analytical chemistry
*Gas chromatography*
Headspace was prepared by gently backfilling sample bottles with 20–30 mL helium (He; 99,9999%)
into the closed bottle while removing a corresponding volume of water. Care was taken to control the
headspace pressure within 5% of ambient and a slight He overpressure was released before
equilibration. The bottles were shaken horizontally at 150 rpm for 1 h to equilibrate gases between
sample and headspace. The temperature during shaking was recorded by a data logger. Immediately
after shaking, the bottles were placed in an autosampler (GC-Pal, CTC, Switzerland) coupled to a gas
chromatograph (GC) with He back-flushing (Model 7890A, Agilent, Santa Clara, CA, US).
Headspace gas was sampled (approx. 2 mL) by a hypodermic needle connected to a peristaltic pump
(Gilson Minipuls 3), which connected the autosampler with the 250 µL heated sampling loop of the
GC.
The GC was equipped with a 20-m wide-bore (0.53 mm) Poraplot Q column for separation of CH$_4$,
CO$_2$ and N$_2$O and a 60 m wide-bore Molsieve 5Å PLOT column for separation of O$_2$ and N$_2$, both
operated at 38°C and with He as carrier gas. N$_2$O and CH$_4$ were measured with an electron capture
detector run at 375°C with Ar/CH$_4$ (80/20) as makeup gas, and a flame ionization detector,
respectively. CO$_2$, O$_2$, and N$_2$ were measured with a thermal conductivity detector (TCD). Certified
standards of CO$_2$, N$_2$O, and CH$_4$ in He were used for calibration (AGA, Germany), whereas air was
used for calibrating O$_2$ and N$_2$. The analytical error for all gases was lower than 2%. For the Lake
Lundebyvannet time series, CO$_2$ was separated from other gases using the 20 m wide-bore (0.53 mm)
Poraplot Q column while the other gases were not measured.
The results from gas chromatography give the relative concentration of dissolved gases (in ppm) in
the headspace in equilibrium with the water. For the lab experiment with Svartkulp samples (section
2.1), the concentration of dissolved gases in the water at equilibrium with the headspace were
calculated from the temperature corrected Henry constant in water using Carroll, Slupsky and Mather
(1991) for CO$_2$, Weiss and Price (1980) for N$_2$O, Yamamoto, Alcauskas and Crozier (1976) for CH$_4$,
Millero, Huang and Laferiere (2002) for O$_2$, Hamme and Emerson (2004) for N$_2$. For the Lake
Lundebyvannet time series (section 2.2), the concentration of $CO_2$ in the water samples were
determined using temperature-dependent Henry's law constants given by Wilhelm, Battino and
Wilcock (1977). The quantities of gases in the headspace and water were summed to find the
concentrations and partial pressures of dissolved gases from the water collected in the field as follows:
$$[gas] = \frac{p_{gas}HV_{water} + \frac{p_{gas}V_{headspace}}{RT}}{V_{water}} \qquad \text{(Eq. 1)}$$
where [gas] is the gas aqueous concentration, $p_{gas}$ is the gas partial pressure, $H$ is the Henry constant,
$V_{water}$ is the volume of water sample during headspace equilibration, $V_{headspace}$ is the headspace gas
volume during equilibration, $R$ is the gas constant and $T$ the temperature during headspace
equilibration (recorded during shaking). The calculations were similar to Yang *et al.* (2015).

*DIC analyses*
DIC analysis was performed for the Lake Lundebyvannet time series using a Shimadzu TOC-V CPN
(Oslo, Norway) instrument equipped with a non-dispersive infrared (NDIR) detector with $O_2$ as a
carrier gas at a flow rate of 100 mL min⁻¹. Two to three replicate measurements were run per sample.
The system was calibrated using a freshly prepared solution containing different concentrations of
NaHCO₃ and Na₂CO₃ and standards were measured in between each 6ᵗʰ sample. $CO_2$ concentrations
in water samples ($[CO_2]$) were calculated on the bases of temperature, pH and DIC concentrations as
follows (Rohrlack *et al.*, 2020):
$$[CO_2] = \frac{[H^+]^2 C_T}{Z} \qquad \text{(Eq. 2)}$$
where $[H^+]$ is the proton concentration ($10^{-pH}$), $C_T$ is the dissolved inorganic carbon concentration
and $Z$ is given by:
$$Z = [H^+]^2 + K_1[H^+] + K_1K_2 \qquad \text{(Eq. 3)}$$
where $K_1$ and $K_2$ are the first and second carbonic acid dissociation constant adjusted for temperature
(pK₁ = 6.41 and pK₂ = 10.33 at 25°C; Stumm & Morgan, 1996).

2.4. Data analysis
*pCO₂ and saturation deficit*
Lake Lundebyvannet $CO_2$ concentrations provided by GC and DIC analyses were converted to pCO₂
(in µatm) as follows:
$$pCO_2 = \frac{[CO_2]}{0.987 \times K_H\, P_{atm}} \qquad \text{(Eq. 4)}$$

where $K_H$ is Henry constant for CO$_2$ adjusted for in-situ water temperature (Stumm & Morgan, 1996)
and $P_{atm}$ is the atmospheric pressure in bar approximated by:
$$P_{atm} = (1013 - 0.1 \times altitude) \times 0.001 \qquad \text{(Eq. 5)}$$

where $altitude$ is the altitude above sea level of Lake Lundebyvannet (158 m). Finally, the CO$_2$
saturation deficit ($Sat_{CO_2}$ in µatm) was given by
$$Sat_{CO_2} = pCO_2 - [CO_2]_{air} \qquad \text{(Eq. 6)}$$

where $[CO_2]_{air}$ is the pCO$_2$ in the air (416 µatm for 2020 in Southern Norway retrieved from EBAS
database; NILU, 2022; Tørseth et al., 2012). $Sat_{CO_2}$ gives the direction of CO$_2$ flux at the water-
atmopshere interface, and its product with gas transfer velocity determine the CO$_2$ flux at the water-
atmosphere interface, i.e., whether lake ecosystems are sink ($Sat_{CO_2} < 0$) or source ($Sat_{CO_2} > 0$) of
atmospheric CO$_2$.

*Statistical analyses*
The effect of storage time and treatment on five dissolved gases (O$_2$, N$_2$, CO$_2$, CH$_4$, N$_2$O) from the
Lake Svartkulp samples was tested with a two-way ANOVA at an alpha level adapted using the
Bonferroni correction for multiple testing, i.e., α=0.05/5=0.01. To evaluate the impact of Hg fixation
on Lake Lundebyvannet samples, $[CO_2]$ values determined by headspace equilibration and GC
analysis of HgCl$_2$-fixed samples were compared with those calculated from DIC measurements of
unfixed samples with a paired t-test.
A regression analysis was performed to describe the overestimation of CO$_2$ concentrations caused by
HgCl$_2$ fixation in Lake Lundebyvannet samples as a function of pH. The total CO$_2$ concentration in
the HgCl$_2$-fixed samples ($[CO_2]_{HgCl_2}$) can be expressed as:
$$[CO_2]_{HgCl_2} = [CO_2]_i + [CO_2]_{ex} \qquad \text{(Eq. 7)}$$

where $[CO_2]_i$ is the initial CO$_2$ concentration prior to HgCl$_2$ fixation, i.e., CO$_2$ concentration in the
unfixed samples, and $[CO_2]_{ex}$ is the excess CO$_2$ concentration caused by a decrease in pH following
HgCl$_2$ fixation. The relative CO$_2$ overestimation ($E$ in %) is given by:
$$E = \frac{[CO_2]_{HgCl_2} - [CO_2]_i}{[CO_2]_i} = \frac{[CO_2]_{ex}}{[CO_2]_i} \qquad \text{(Eq. 8)}$$

The impact of pH (or $[H^+]$) on $E$ was mathematically described by running a regression analysis
using MATLAB®. The *fminsearch* MATLAB function from the Optimization toolbox was used to
find the minimum sum of squared residuals (SSR) for functions of the form of: $E = A/[H^+]$ or $E =$
$A \times 10^{-B \times pH}$. For each optimal solution, the root-mean-square error (RMSE) and coefficient of
determination ($R^2$) were calculated against observed values of $E$, i.e., values of $E$ determined
empirically from observed $[CO_2]_i$ and $[CO_2]_{ex}$.

*Chemical speciation, saturation-index calculations, and prediction of CO₂ overestimation*
The speciation of solutes and saturation index values (SI) of selected minerals were calculated with
the program PHREEQC developed by the USGS (Parkhurst & Appelo, 2013), neglecting the effect of
dissolved organic matter. This was used to assess the impact of the addition of preservative on pH and
shifting the carbonate equilibrium as well as dissolved inorganic carbon losses due to carbonate
mineral precipitation. PHREEQC is commonly used to calculate the speciation of inorganic carbon,
the SI of carbonate minerals and to help estimate the fate of inorganic carbon in carbon cycling
studies (Atekwana et al. 2016; Clayer et al. 2016; Klaus 2023). For each PHREEQC simulation, two
files, respectively the database (with input reactions) and input files, were used to define the
thermodynamic model and the type of calculations to perform. The database of MINTEQA2 (e.g.,
*minteq.dat*, Allison et al., 1991) was used to describe the chemical system because it includes, inter
alia, reactions and constants for Ag, Cu and Hg complexation with Cl, NO₃ and carbonates.
Three PHREEQC simulations were run representing the addition of each preservative solution to
sample water from Lake Svartkulp. The input files described the composition of two aqueous
solutions: (i) the preservative solution assumed to contain only the preservative (i.e., HgCl₂ solution)
and (ii) sample water from Lake Svartkulp with observed major element concentrations (pH, Al, Ca,
Cl, Cu, Fe, Mg, Mn, N as nitrate, K, Na, S as sulfate, Zn; Tab. S1) and Hg and Ag natural
concentration assumed to be $10^{-5}$ mg/L. The output file provided the activities of the various solutes in
the preserved samples, i.e., simulating the mixing of 120 mL of lake water with 240 µL of the AgNO₃,
CuCl₂ and HgCl₂ preservative solutions, as described in section 2.1. This procedure allows to estimate
the pH of the preserved samples as well as SI for various mineral phases. The SI is calculated by
PHREEQC comparing the chemical activities of the dissolved ions of a mineral (ion activity product,
IAP) with their solubility product (Ks). When SI > 1, precipitation is thermodynamically favourable.
Note however that PHREEQC does not give information about precipitation kinetics.
Similarly, PHREEQC was used to estimate the decrease in pH caused by adding 150 µL of a half-
saturated HgCl₂ solution to Lake Lundebyvannet samples prior to GC analyses, as described in
section 2.2. In absence of data on the chemical composition of Lake Lundebyvannet, we assumed that
it had the same composition as Lake Svartkulp water samples. This assumption is supported by the
fact that waters from both lakes have circumneutral pH, low ionic strength (poor buffering capacity)
and high DOC concentration and would therefore behave similarly in presence of acids. Briefly, for
each 0.1 pH value between pH of 5.4 and 7.3, the carbonate alkalinity was first adjusted by increasing
HCO$_3$ concentrations in the input files for PHREEQC to confirm that the water was at equilibrium at
the given pH value. Then, the effect of adding 150µL of a half-saturated HgCl$_2$ solution was
simulated as described above for Lake Svartkulp. Knowing the new equilibrated pH, after addition of
HgCl$_2$, the overestimation of CO$_2$ concentration in Hg-fixed samples relative to unfixed samples ($E$,
described in Eq. 8 above) can be predicted as described below.
Adapting Eq. (2), we obtain:

$$[CO_2]_{HgCl_2} = \frac{[H^+]^2_{HgCl_2} \, C_T}{Z_{HgCl_2}} \qquad \text{(Eq. 9)}$$

and

$$[CO_2]_i = \frac{[H^+]^2_i \, C_T}{Z_i} \qquad \text{(Eq. 10)}$$

where $[H^+]_i$ is the proton concentration measured in the initial water samples prior to HgCl$_2$ fixation,
and $[H^+]_{HgCl_2}$ is the proton concentration estimated by PHREEQC following HgCl$_2$ fixation, and
similarly for $Z_i$ and $Z_{HgCl_2}$ from Eq. (3). Combining Eqs. (7), (9) and (10) we obtain:

$$[CO_2]_{ex} = C_T \left( \frac{[H^+]^2_{HgCl_2}}{Z_{HgCl_2}} - \frac{[H^+]^2_i}{Z_i} \right) \qquad \text{(Eq. 11)}$$

Hence:

$$E = \frac{[CO_2]_{ex}}{[CO_2]_i} = \frac{\left( \frac{[H^+]^2_{HgCl_2}}{Z_{HgCl_2}} - \frac{[H^+]^2_i}{Z_i} \right)}{\frac{[H^+]^2_i}{Z_i}} \qquad \text{(Eq. 12)}$$

Alternatively, $E$ can also simply be predicted based on the carbonic acid dissociation:

$$CO_2 + H_2O \overset{K_1}{\Leftrightarrow} HCO_3^- + H^+ \qquad \text{(Reaction 1)}$$

At equilibrium, we have:

$$K_1 = \frac{[HCO_3^-][H^+]}{[CO_2]} \qquad \text{(Eq. 13)}$$

When pH is decreased upon addition of HgCl₂, a fraction ($\alpha$) of the initial bicarbonate concentration
$[HCO_3^-]_i$ is turned into $CO_2$. This fraction, expressed as $[CO_2]_{ex}$ in Eq. (7) above, can be estimated
with Eq. 13 as follows:

$$[CO_2]_{ex} = \alpha[HCO_3^-]_i = \frac{\alpha K_1 [CO_2]_i}{[H^+]_i} \qquad \text{(Eq. 14)}$$

Introducing the expression of $[CO_2]_{ex}$ from Eq. 14 into Eq. 8 yields:

$$\frac{[CO_2]_{ex}}{[CO_2]_i} = E = \frac{\alpha K_1}{[H^+]_i} \qquad \text{(Eq. 15)}$$

When the decrease in pH, or acidification, is greater than the buffering capacity of the water: $\alpha = 1$.
The value of $\alpha$ cannot exceed 1 because the amount of CO₂ produced by a decrease in pH cannot
exceed the amount of $HCO_3^-$ initially present. In all the other cases, we have: $\alpha < 1$. For both
predictions of $E$, i.e., with Eqs. 12 and 15, the root-mean-square error (RMSE) and coefficient of
determination ($R^2$) were calculated.
Finally, additional sources of CO₂ overestimation were investigated by analysing the residuals of the
model described by Eq. 12, i.e., the difference between $E$ predicted with Eq. 12 and $E$ determined
empirically with Eq. 8. Briefly, residuals were plotted against pH and *in situ* temperature. Residuals
were separated in two groups based on the empirical value of $[HCO_3^-]_i - [CO_2]_{ex}$, i.e., the first group
had values of $[HCO_3^-]_i - [CO_2]_{ex} \geq a$ while the second group had values of $[HCO_3^-]_i - [CO_2]_{ex} \leq$
$-a$ where different values for $a$ were used: 20, 10 or 5 µM. The justification for separating residuals
in two groups is that: (i) the first group represents samples for which bicarbonate alkalinity in the
original sample is, as expected, higher than CO₂ overestimation after HgCl₂-fixation, while (ii) the
second group represents samples for which bicarbonate alkalinity is not sufficient to explain CO₂
overestimation after HgCl₂-fixation.

*CO₂ diffusion fluxes from Lake Lundebyvannet*
The diffusive flux of CO₂ ($F_{CO_2}$ in mol m⁻² d⁻¹) from Lake Lundebyvannet surface water was
estimated according to:

$$F_{CO_2} = \frac{k_{CO_2}([CO_2] - [CO_2]_{eq})}{1000} \qquad \text{(Eq. 16)}$$

where $k_{CO_2}$ is the CO₂ transfer velocity in m d⁻¹, $[CO_2]$ is the surface water CO₂ concentration (µM),
and 1000 is a factor to ensure consistency in the units and $[CO_2]_{eq}$ is the theoretical water CO₂
concentration (µM) in equilibrium with atmospheric CO₂ concentration calculated with Eq. (3) and
pCO₂ of 416 µatm (see above).
The CO$_2$ transfer velocity ($k_{CO_2}$) was estimated as follows (Vachon & Prairie, 2013):
$$k_{CO_2} = k_{600} \left( \frac{600}{Sc_{CO_2}} \right)^{-n} \qquad \text{(Eq. 17)}$$

where $k_{600}$ is the gas transfer velocity (m d$^{-1}$) estimated from empirical wind-based models and $Sc_{CO_2}$
is the CO$_2$ Schmidt number for in situ water temperature (unitless; Wanninkhof, 2014). We used n
values of 0.5 or 2/3 when wind speed was below or above 3.7 m s$^{-1}$, respectively (Guérin et al., 2007).
Empirical $k_{600}$ models included those from Cole & Caraco (1998; $k_{600} = 2.07 + 0.215 U_{10}^{1.7}$),
Vachon & Prairie (2013; $k_{600} = 2.51 + 1.48 U_{10} + 0.39 U_{10} \log_{10} LA$) and Crusius & Wanninkhof
(2003; power model: $k_{600} = 0.228 U_{10}^{2.2} + 0.168$ in cm h$^{-1}$). $U_{10}$ and $LA$ refer to mean wind speed at
10 m in m s$^{-1}$ and lake area in km$^2$, respectively. Sub-hourly $U_{10}$ data for 2020 was retrieved from a
weather station of the Norwegian Meteorological Institute located 1.5 km west of Lake
Lundebyvannet (station name: E18 Melleby; ID: SN 3480; 59.546 N, 11.4535E) using the Frost
application programming interface (*Frost API*, 2022). Daily, monthly, and yearly (only covering the
ice-free season: April-November) $F_{CO_2}$ was estimated using Eq. (12). Daily [CO$_2$] was interpolated
from weekly data using a modified Akima spline (makima spline in Matlab® based on Akima, 1974).
This interpolation method is known to avoid excessive local undulations.

**3. Results**

3.1. Effects of preservatives and storage time on dissolved gases
In the unfixed samples from Lake Svartkulp, the concentration of O$_2$ declined while CO$_2$ increased
over time in a close to 1:1 molar ratio, likely reflecting the effect of microbial respiration activity and
mineralisation of organic matter (Fig. 2, Tab. S2). Concentration of O$_2$ in the unfixed samples
decreased from near 300 to below 200 µM (Fig. 2). In the presence of inhibitors, O$_2$ concentrations
tended to be slightly higher at t=24h and remained constant or declined only slightly over time to
generally remain at or above saturation (280 to 300 µM). Thus, the inhibitors were effective in
reducing oxic respiration.
The concentration of CO$_2$ in the presence of AgNO$_3$ at t = 24h was not significantly different to the
unfixed at t = 0 (Fig 2; paired t-test, *P*>0.1). At t = 24h, CO$_2$ concentrations were however much
higher in the presence of HgCl$_2$ (135 µM) or CuCl$_2$ (131 µM) than in the unfixed (89 µM; Fig 2, Tab.
S2). The CO$_2$ further increases from 130 µM to ~160 µM after 3 weeks in both sample sets preserved
with HgCl$_2$ and CuCl$_2$ while a decrease in O$_2$ is less pronounced  for samples fixed with CuCl$_2$ and
completely absent for samples fixed with HgCl$_2$. Overall, the addition of HgCl$_2$ or CuCl$_2$ following
sampling increased CO$_2$ concentrations by 47% after 24h compared to the unfixed and caused further
changes over the three-month storage time, while preservation with AgNO$_3$ yielded CO$_2$
concentrations consistent with the unfixed and caused negligible changes over time (Fig. 2; paired t-
test, P>0.1).
The concentration of CH$_4$ across all samples ranged between 0.017 and 0.377 µM (Fig. 2), as
expected two orders of magnitude smaller than CO$_2$. At t = 24h, the concentration of CH$_4$ was over
0.2 µM in the presence of inhibitors while it was below saturation in the unfixed (0.03 µM; Fig. 2).
CH$_4$ oversaturation in the preserved samples persisted after three weeks and three months of storage
and CH$_4$ concentration remained unchanged (Fig. 2, Tab. S2).
The concentration of N$_2$O ranged between 9.8 and 12.7 nM with only samples preserved with AgNO$_3$
showing negligible changes over time (Fig. 2; paired t-test, P>0.1). All the other samples showed
consistent patterns with storage time. N$_2$O concentrations initially increased within the first 3 weeks,
followed by a decrease after 3 months.
The changes in N$_2$ were likely within handling and analytical errors and not different in the presence
or absence of inhibitors (Fig. 2; Tab. S2; paired t-test, P>0.1).

3.2. Effects of preservatives on pH
In the samples amended with ultrapure water or AgNO$_3$, the pH did not show any significant changes
after 2h or 24h. In contrast, both groups with HgCl$_2$ and CuCl$_2$ treatments show significant decreases
of pH after 2h, -0.12 and -0.19, respectively, and 24h, -0.16 and -0.21, respectively. In addition, they
showed a significant decrease in pH from 2h to 24h. Samples amended with CuCl$_2$ show the strongest
decrease in pH.
3.3. Contrasting impacts of HgCl$_2$, CuCl$_2$ and AgNO$_3$ on dissolved CO$_2$ estimation revealed by
chemical speciation modelling
The PHREEQC simulation of unpreserved samples, based on concentrations of all major elements
(Tab. S1), predicted a pH of 6.72 (Tab. 2) which is very close to the measured pH of 6.73 (Tab. S1).
This suggests that chemical information provided to PHREEQC is likely sufficient to describe the
system, without having to invoke more complex reactions with dissolved organic matter. The addition
of HgCl$_2$ and CuCl$_2$ both caused a significant decrease in pH to 6.40 and 6.45, respectively (Tab. 2)
which is similar to the decrease observed at the end of the 24h short term incubation (Fig. 2).
In absence of preservatives, none of the common carbonate minerals, including calcite, were
associated with a saturation index higher than 1, i.e., dissolution was thermodynamically favourable
for all these minerals and no DIC loss was expected (Tab. 2). However, upon addition of HgCl$_2$ or
CuCl$_2$, some carbonate minerals, e.g., HgCO$_3$ or malachite and azurite, respectively, were expected to
spontaneously precipitate given their relatively high saturation index values.
3.3. Effects of $HgCl_2$ on dissolved $CO_2$ concentration under a range of pH
$CO_2$ concentrations in unfixed water samples from Lake Lundebyvannet were significantly lower than
in the $HgCl_2$-fixed samples (mean difference: 52 µM; paired t-test; P<0.0001; Tab. 3). Fixation with
$HgCl_2$ caused a general overestimation of $CO_2$ concentration and the saturation deficit (Fig. 4), thus
missing out events of $CO_2$ influx (carbon sink) under high photosynthesis activity (high pH; Fig. 4).
In parallel, PHREEQC predicted a decrease of 0.6 to 1.8 units of pH related to $HgCl_2$ addition (Fig.
S1).
The pH value of water samples from Lake Lundebyvannet varied between 5.4 and 7.3 (Fig 4 and 5),
mainly due to marked variations in phytoplankton photosynthetic activity (Rohrlack et al., 2020). The
relative overestimation of $CO_2$ ($E$) follows an exponential increase with pH and is well reproduced by
a simple exponential function ($2.56 \times 10^{-5} \times 10^{1.015 \times pH}$, RMSE=44%, $R^2$=0.81, p<0.0001; Fig. 5).

## 4. Discussion

Prior to using dissolved gas concentrations in freshwater to estimate the magnitude of biological
aquatic processes such as photosynthesis and oxic respiration, denitrification and methanogenesis, we
must ensure that biological activity between sampling and laboratory analyses was efficiently
inhibited without significant impacts on the sample's chemistry. Here we report a unique dataset on
the impact of three preservatives on water samples from a typical low-ionic strength, unproductive
boreal lake to inform on potential risks of mis-estimation of dissolved gas concentrations. We further
show, using $CO_2$ concentration data from a typical productive boreal lake, that using $HgCl_2$ can lead
to negligence of the role of photosynthesis in lake C cycling.
4.1 Best preservative for the determination of dissolved gas concentrations
Given that none of the four treatments (unfixed, $HgCl_2$, $CuCl_2$ or $AgNO_3$) applied to Lake Svartkulp
water samples during the 3-month incubation offer an independent control, a first challenge is to
determine which of the treatment represent the most realistic dissolved gas concentrations close to
real condition. For $CO_2$ and $O_2$, a few studies have used unfixed samples (only preserved dark at
+4°C) up to 48h after sampling to determine $CO_2$ or DIC concentrations (e.g., Sobek et al. 2003,
Kokic et al., 2015). So, the $CO_2$ and $O_2$ concentrations in the unfixed samples collected after 24h
incubation are the most representative of the initial real concentrations. Biological activity might have
had an impact, but this is likely negligible over the first 24h. In addition, the fact that the $CO_2$ and $O_2$
concentrations in the samples fixed with $AgNO_3$ after 24h, three weeks and three months are equal to
those from unfixed samples after 24h (Fig. 2) confirms that the unfixed samples after 24h can be used
as a control. In fact, only samples fixed with $AgNO_3$ are trustful given the expected toxicity of Ag, the
absence of impact on pH (Fig. 3), and unchanged concentrations over the three-month experiment for
all gases. Similarly, $N_2O$ and $N_2$ concentrations in the unfixed samples after 24h can be used as
control. However, for $CH_4$, Fig. 2 shows that already after 24h, the $CH_4$ concentration in the unfixed
samples is below atmospheric saturation while it is consistently much higher in all three sets of fixed
samples. Boreal lakes are typically over saturated with respect to $CH_4$ (Valiente et al., 2022) and it is
very unlikely that $CH_4$ could have been produced in lake water incubated under high concentration of
oxygen and toxic preservatives. Hence, unfixed samples do not represent real $CH_4$ concentrations.
These observations are all consistent with the fact that the three preservatives were effective in
preserving $CH_4$ from oxidation. Even over 24h, preservatives need to be added to oxic water samples
to preserve $CH_4$ from oxidation. In fact, oxic methanotrophy typically show rates in the order of µM
day$^{-1}$ (Thottathil et al., 2019; van Grinsven et al., 2021). Hence, a $CH_4$ consumption of 0.3 µM within
24h in the unfixed water samples is realistic (Fig. 2).
In summary, preservation with $AgNO_3$ is the only method that offered robust determination of all five
dissolved gases with negligible changes in concentration over time.
4.2 Risks of mis-estimating dissolved gas concentration with $HgCl_2$ and $CuCl_2$ preservation
Both sets of samples preserved with either $HgCl_2$ and $CuCl_2$ showed $CO_2$ concentrations that were
much higher than the unfixed (after 24h) or the $AgNO_3$-fixed samples. This is due to an acidification
of the poorly buffered (alkalinity 127 µM) and near neutral water (pH=6.73), shifting the carbonate
equilibrium from $HCO_3$ to $CO_2$ as also shown by Borges et al. (2019). In fact, a rapid decrease in pH
was observed upon $HgCl_2$ and $CuCl_2$ treatments (Fig. 3). The increase of $CO_2$ from about 130 µM to
~160 µM after 3 weeks in both sample sets preserved with $HgCl_2$ and $CuCl_2$ is not mirrored by a
similar decrease in $O_2$ (Fig. 2). This suggests that oxic respiration is not the main source for this
additional 30 µM of $CO_2$ but rather points towards additional acidification of the samples caused, e.g.,
by kinetically controlled complexation of $Hg^{2+}$ with dissolved organic matter (Miller et al., 2009). In
fact, the relatively slow complexation of $Hg^{2+}$ with organic thiol groups can release two protons
(Skyllberg, 2008) and up to three, with some participation of a third weak-acid group (Khwaja et al.,
2006). The transient nature of acidification caused by $HgCl_2$ and $CuCl_2$ is also evident in the pH
impacts showing higher acidification after 24h than after 2h incubation (Fig. 3). The following
decrease in $CO_2$ after 3 months (down to ~145 µM) points to other processes. The precipitation of Hg
and Cu carbonates, given their high saturation index values (Tab. 2), would be consistent with the
decrease in $CO_2$ concentrations observed between three weeks and three months. Calcite precipitation
is typically observed in supersaturated solutions within 48h (Kim et al., 2020). Hence, it is realistic to
consider that Hg and Cu carbonate precipitation influenced the $CO_2$ concentration within the
preserved samples over the three months of storage time. Impacts of Hg or Cu carbonate precipitation
is not evident after three weeks likely because of slow but persistent $CO_2$ production in presence of
HgCl$_2$ and CuCl$_2$ related to acidification as described above (Fig. 2). However, after three weeks, this
production likely weakens and is counterbalanced by increasing carbonate precipitation.
Overall, the addition of HgCl$_2$ or CuCl$_2$ following sampling increased CO$_2$ concentrations by 47%
within the first 24h compared to the unfixed consistent with the -0.16 to -0.21 pH-unit acidification
observed over the same time in the pH incubation experiment (Fig. 3) and the pH estimated with
PHREEQC without the interaction with dissolved organic matter (Tab. 2). In fact, introducing pH and
CO$_2$ concentration values of 6.40–6.45 and 130 µM, respectively, for the samples preserved with
HgCl$_2$ and CuCl$_2$ into Eqs. 1 and 2 yields DIC concentrations (C$_T$) of about 270 µM at t=24h. These
DIC concentrations are almost equal to those calculated for the unfixed samples and those preserved
with AgNO$_3$ at t = 24h, i.e., with a pH of 6.73 and CO$_2$ concentration of 88 µM. Interestingly, the
concentration of CO$_2$ in the samples preserved with HgCl$_2$ and CuCl$_2$ continues to increase up to ~160
µM after 3 weeks. Given that oxic respiration is inhibited (Fig. 2), this additional CO$_2$ is believed to
originate from progressive release of protons following relatively slow complexation of Hg$^{2+}$ with
dissolved organic matter (Khwaja et al., 2006; Miller et al., 2009; Skyllberg, 2008). Note however
that this process could not be predicted with PHREEQC given that it neglected the effect of dissolved
organic matter.
Unlike the AgNO$_3$-fixed samples, all the other samples showed an initial increase in N$_2$O
concentration from 24h to 3 weeks, followed by a decrease from three weeks to 3 months. Similar
patterns of net N$_2$O production followed by net consumption were also reported in short-term
incubations of seawater from the high latitude Atlantic Ocean, although over much shorter timescales,
i.e., 48 and 96h (Rees et al., 2021). The large difference in kinetics between the latter experiment
(Rees et al., 2021) and our incubation might be attributable to differences in incubation temperature
where the seawater from the high latitude Atlantic Ocean was incubated at ambient temperatures
while our samples were kept at +4°C. Other difference in the experimental setup might have also
played a role. The lack of inhibition of N$_2$O production and consumption in the samples preserved
with HgCl$_2$ and CuCl$_2$ can be attributed to the fact that N$_2$O production tends to increase under
increasing acidic conditions (Knowles, 1982; Mørkved et al., 2007; Seitzinger, 1988). In fact, the
mole fraction of N$_2$O produced during denitrification increases compared to N$_2$ as pH decreases
(Knowles, 1982).

4.3 Using PHREEQC to estimate acidification caused by HgCl$_2$ in samples from Lake Lundebyvannet
As for the samples from Lake Svartkulp as described above, the overestimation of CO$_2$ concentration
in the samples from Lake Lundebyvannet fixed with HgCl$_2$ (161 µM added; Fig. 4) likely stems from
the acidification shifting the carbonate equilibrium from bicarbonate to CO$_2$. In fact, PHREEQC
predicted a decrease of 0.6 to 1.8 units of pH related to HgCl$_2$ addition in these samples (Fig. S1).
The relative overestimation of $CO_2$ ($E$ in Fig. 5) followed a typical exponential increase reflecting the
decrease in absolute $CO_2$ concentration with increasing pH (Stumm & Morgan, 1981) caused here by
phytoplankton photosynthesis. In fact, the exponential increase in $CO_2$ overestimation is easily
predicted by Eq. (9) with an equivalent level of accuracy as the optimized exponential function (Fig.
5). Consistently, the relative overestimation of $CO_2$ ($E$) shows an inverse decrease with $[H^+]$ that is
well reproduced by a simple inverse function ($3.25 \times 10^{-5}/[H^+]$; RMSE=44%, $R^2$=0.81, p<0.0001;
Fig. 5) and predicted by Eq. (15), with an α value of 1. Combining Eqs. 8 and 15 and solving it with
pH values estimated from PHREEQC (Fig. S1) for α yields values ranging between 0.72 and 0.89
with an average of 0.85. Unexpectedly, this average α value is almost equal to the ratio of the inverse
function coefficient and $K_1$, i.e., $\frac{3.25 \times 10^{-5}}{K_1} = 0.87$. Hence, the relative overestimation of $CO_2$ ($E$)
caused by HgCl₂ fixation is easily predicted by the change in bicarbonate equilibrium knowing the
proton release from HgCl₂ addition, here estimated with PHREEQC.
Hence, PHREEQC can be used to predict decrease in pH caused by HgCl₂ fixation, if sufficient
knowledge is gathered on the ionic water composition. Proton release during HgCl₂ fixation can be
represented by the following reaction:
$$HgCl_2 + 2H_2O = Hg(OH)_2 + 2H^+ + 2Cl^- \qquad \text{(Reaction 2)}$$
From reaction 2, it becomes evident that the initial concentration of chloride in the water samples will
likely limit HgCl₂ dissociation and proton release. This is a likely mechanism occurring in seawater
where HgCl₂ has been shown to cause a decrease in pH, although at a negligible level with a
maximum decrease in pH of -0.01 (Chou et al., 2016).
Figure 3 shows that a range of water samples were associated with a relative $CO_2$ overestimation ($E$)
that substantially deviated from the overestimation predicted with Eq. 12 (red and blue symbols in
Fig. 5). In fact, some samples had a higher initial bicarbonate content ($[HCO_3^-]_i$) than the excess $CO_2$
concentration ($[CO_2]_{ex}$), while other showed the opposite. The former case (blue symbols in Fig. 5)
can easily be explained by a higher buffering capacity of the sampled water, i.e., a higher pH after
HgCl₂-fixation than that predicted by PHREEQC related to a different water composition. Indeed, the
concentration of major elements in the water from Lake Lundebyvannet may vary significantly over
time, and in absence of data, we considered that the water composition, except for DIC, pH and
HgCl₂, was constant over time. By contrast, samples associated with $[CO_2]_{ex}$ being larger than
$[HCO_3^-]_i$ are more enigmatic. In order to shed light on possible explanations, we visually inspected
trends between empirical deviations from predictions, i.e., residuals, and *in situ* temperature or pH.
Absolute values of residuals showed a progressive increase with pH and *in situ* temperature which is
in agreement with decreasing precision of the headspace method with increasing temperature and pH
(Koschorreck et al., 2021). In fact, $CO_2$ is less soluble at higher temperature, hence more gas can
evade during sampling, and thus the error increases with *in situ* temperature. In addition, at higher pH,
CO₂ concentration decreases and consequently the absolute error on CO₂ quantification becomes
larger relative to measured CO₂ concentration. Interestingly, many of the high residual values were
not evenly distributed across the year, nor across the summer and were rather associated with only a
few specific sampling events during summer (Fig. S2). This suggests that degassing could have
occurred due to high ambient temperature in the field. Water associated with $[CO_2]_{ex}$ being larger
than $[HCO_3^-]_i$ (red symbols in Fig. 5 and S4) could have been subject to a larger degassing in the
samples collected for DIC analysis than the samples for GC analysis. On the other hand, degassing
was likely larger for samples for GC analysis than for DIC analysis for water associated with
$[HCO_3^-]_i$ being larger than $[CO_2]_{ex}$ (blue symbols in Fig. 5 and Fig. S2). In addition to degassing and
temperature effects, errors in pH measurements can also cause a large misestimation of CO₂
concentration from DIC analysis, and this error increases exponentially with pH following the shift in
carbonate equilibrium. In summary, our analysis is consistent with that of Koschorreck et al. (2021)
showing that errors in the determination of CO₂ concentrations are smaller at lower pH and lower
temperature (Fig. S2).
4.4. Implications for the estimation of lake and reservoir C cycling and recommendations
Using HgCl₂ or CuCl₂ to preserve dissolved gas samples in poorly buffered water samples has large
impacts on CO₂ concentrations with considerable risk of leading to incorrect interpretations. The risk
of mis-estimating CO₂ concentration due to HgCl₂ and CuCl₂ preservation is the highest when pH of
the unfixed water is close to the first carbonic acid dissociation constant (pK₁ = 6.41 at 25°C; Stumm
& Morgan, 1996). It implies that any small shift in pH will have a significant impact in the carbonate
equilibrium between bicarbonate to CO₂. The risk is also the highest in the lowest ionic strength
waters. In that respect, low-ionic strength, slightly acidic to neutral, moderately humic lakes
commonly found in Norway (de Wit et al., 2023), large parts of Sweden (Valina et al. 2014), and
Finland, Atlantic Canada (Houle et al., 2022), Ontario, Québec, and North-East USA (Skjelkvåle and
de Wit 2011; Weyhenmeyer et al., 2019) are the most prone to errors in CO₂ concentrations related to
HgCl₂ or CuCl₂ preservation. Through a preliminary literature search we found several studies from
boreal lakes (Jonsson et al., 2001; Urabe et al., 2011; Yang et al., 2015; Hessen et al., 2017) but also
from circum-neutral pH sub-tropical to tropical aquatic environments (Jeffrey et al., 2018; Webb et
al., 2018; Ray et al., 2021) where preservation with HgCl₂ may have caused biases in the
quantification of CO₂ concentrations as it was the case for samples from the Congo River (Borges et
al., 2019). A significant part of the low-ionic strength boreal lakes become increasingly sensitive to
changes in nutrients with strong impacts on their role in carbon cycling (Myrstener et al., 2022). In
this context, it is crucial to avoid mis-estimation of CO₂ concentrations and thus avoid use of HgCl₂ or
CuCl₂ to ensure a robust understanding of the role of autotrophic processes in lake C cycling. Below
we describe the implications for the lake C budget of Lundebyvannet as an example of a mis-
estimation of the role of photosynthesis in a typical productive boreal lake.
In Lake Lundebyvannet, over the ice-free season, average $CO_2$ concentrations determined following
$HgCl_2$-fixation and GC analysis were 82% higher than those obtained from DIC analyses (Tab. 3; Fig.
6 and S3). $CO_2$ concentrations obtained from $HgCl_2$-fixed samples created the illusion that Lake
Lundebyvannet was a steady net source of $CO_2$ to the atmosphere over the ice-free season with large
$CO_2$ saturation deficit (Fig. 4) while, in reality, the lake switched from being a net source in May, to a
net sink over a few weeks in June, and returning to a net source in July (Fig. 6 and S3). Indeed,
monthly $CO_2$ overestimation related to $HgCl_2$-fixation reached about 300% in June (Tab. 3).
Propagating this overestimation into the estimates of $CO_2$ diffusion fluxes with typical wind-based
models yields overestimation of $CO_2$ fluxes of 108–112% over the ice-free season and up to 2100% in
June (Tab. 3 and S3). Hence, interpreting $CO_2$ data without correcting for $CO_2$ overestimation caused
by $HgCl_2$-fixation leads to negligence of the role of photosynthesis in lake C cycling with major
implications for current and future predictions of lake $CO_2$ emissions.
The use of $HgCl_2$ to preserve water samples prior to dissolved gas analyses is part of the current
guidelines for greenhouse gas measurements in freshwater reservoirs (Machado Damazio et al., 2012;
UNESCO/IHA, 2008, 2010). Hence, there is a risk of overestimating $CO_2$ concentrations and
emissions, in absence of discrete measurement of emissions, from hydropower reservoirs with
consequence on the present and expected greenhouse gas footprint from hydroelectricity. To ensure
precise estimation of greenhouse gas concentration and, possibly, emission from hydropower, the use
of $HgCl_2$ should therefore be discontinued.
**5. Conclusion**
Mercury is a potent neurotoxin for humans and toxic for the environment and its use should be
discouraged, notably following the Minamata convention on mercury, a global treaty ratified by 126
countries (16 December 2020) to protect human health and the environment from the adverse effects
of mercury. This study further questions the use of $HgCl_2$ for preservation of poorly buffered (low
ionic strength) water samples with high DOC concentration for analysis of dissolved gases in the
laboratory. Although $CuCl_2$ is less toxic, it behaved similarly to $HgCl_2$ and cannot be recommend. In
fact, both chlorinated inhibitors caused a significant decrease in pH shifting the carbonate equilibrium
towards $CO_2$ and are also suspected to promote carbonate precipitation over long-term storage. The
only promising inhibitor tested in this study was $AgNO_3$ notably for dissolved $CO_2$, $CH_4$ and $N_2O$.
Silver nitrate is a suitable substitute for $HgCl_2$ in low-ionic strength waters, further tests should be
carried out with a range of inhibitor concentration and more diverse water samples. The use of
chemical inhibitors may not be the best approach. Alternatives exist, such as directly measuring gas
concentrations *in situ* with sensors, or sampling the headspace out in the field, and bringing back gas
samples (e.g., Cole et al., 1994; Karlsson et al., 2013; Kling et al., 1991; Valiente et al., 2022), rather
than water samples, to the lab for gas chromatography analyses. However, care must be taken to know
the exact equilibration temperature (Koschorreck et al., 2021) and to avoid gas exchange with the
atmosphere as well as to use a clean background gas during headspace equilibration which can be
challenging in remote environments under harsh meteorological conditions.
We further advise against interpretation of $CO_2$ concentration data from low ionic strength, circum-
neutral water samples preserved with $HgCl_2$ or $CuCl_2$. The overestimation of $CO_2$ concentration
caused by $HgCl_2$ can mask the effect of photosynthesis on lake carbon balance, creating the illusion
that lakes are net $CO_2$ sources when they are net $CO_2$ sinks. Our analysis from Lake Lundebyvannet
shows that $HgCl_2$ fixation led to an overestimation of the $CO_2$ concentration by a factor of 1.8, on
average, but approaching a factor of 4 during the peak photosynthetic period. An even larger impact is
expected on $CO_2$ diffusive fluxes which were overestimated by a factor of 2 on average and up to a
factor of >20 during peak photosynthesis. Interpreting such data would have underestimated the
current and future role of aquatic photosynthesis.
**Data availability**
All data supporting this study is available at
https://doi.org/10.4211/hs.436be40748a246269102b20211b49762 (Clayer et al., 2024)
**Author contribution**
JET, AK, and TR supervised and PD, KN and FC contributed to the study design. JET, KN and TR
carried out the experiments. PD and TR performed the chemical analyses. JET and FC wrote the first
draft. FC performed the modelling, data, and statistical analyses, and drafted the figures. All co-
authors edited the manuscript.
**Competing interests**
The contact author has declared that none of the authors has any competing interests.
**Acknowledgements**
We are grateful to Benoît Demars for research assistance, coordination, and useful comments and
discussions on an earlier version of this manuscript and to Heleen de Wit for discussions. Research
was funded by NIVA and through the Global Change at Northern Latitude (NoLa) project #200033.

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

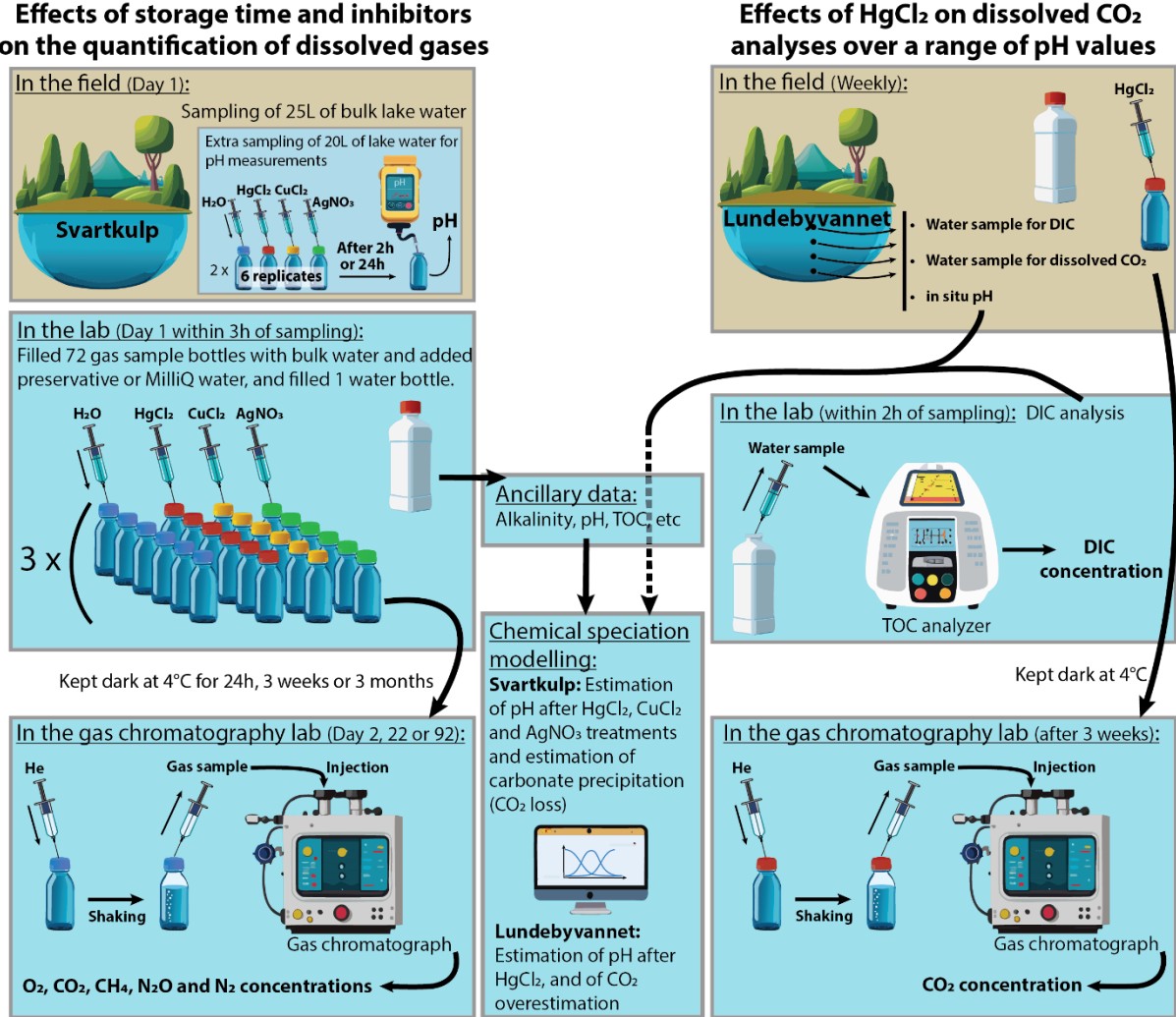


**Fig. 1**. Overview of experimental procedures. Several graphic items in this figure have been generated
with the help of Adobe® Firefly™ Artificial Intelligence generator.

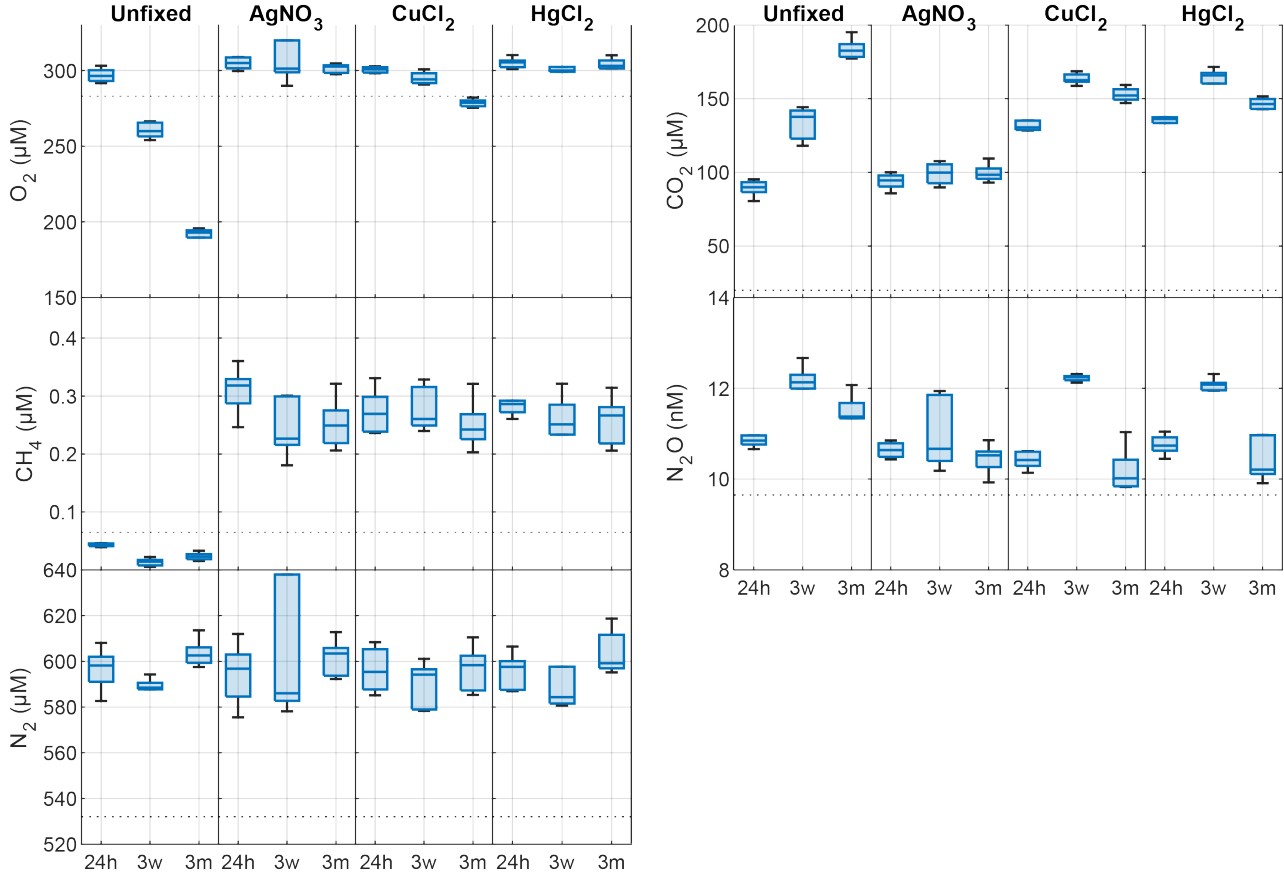

**Fig 2**. Changes in dissolved $O_2$, $CO_2$, $CH_4$, $N_2O$ and $N_2$ concentrations (nM or µM) in the absence (unfixed) and presence of different preservatives ($AgNO_3$, $CuCl_2$, $HgCl_2$) at three times (24h, 24h after incubation start; 3w, three weeks after collection; 3m, three months after collection). The horizontal dotted line is the saturated gas concentration corresponding to 100% gas saturation at *in situ* lake temperature. Box plots show the median, 25th and 75th percentiles and the whiskers display minimum and maximum.

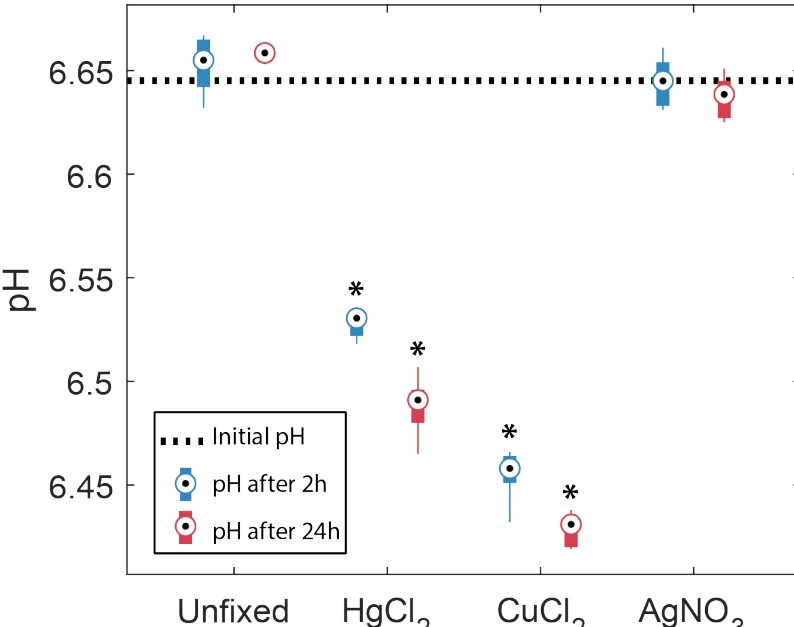

1020

**Fig 3**. Observed changes in pH in the absence (unfixed) and presence of different preservatives (AgNO₃, CuCl₂, HgCl₂) at two times, 2h and 24h after the start of the incubation. The horizontal dotted line represents the initial pH of the bulk water sample. Box plots show the median, 25th and 75th percentiles and the whiskers display minimum and maximum of the 6 replicates. Stars indicate groups that are significantly different from each other and from the initial pH (two-way ANOVA).

1026

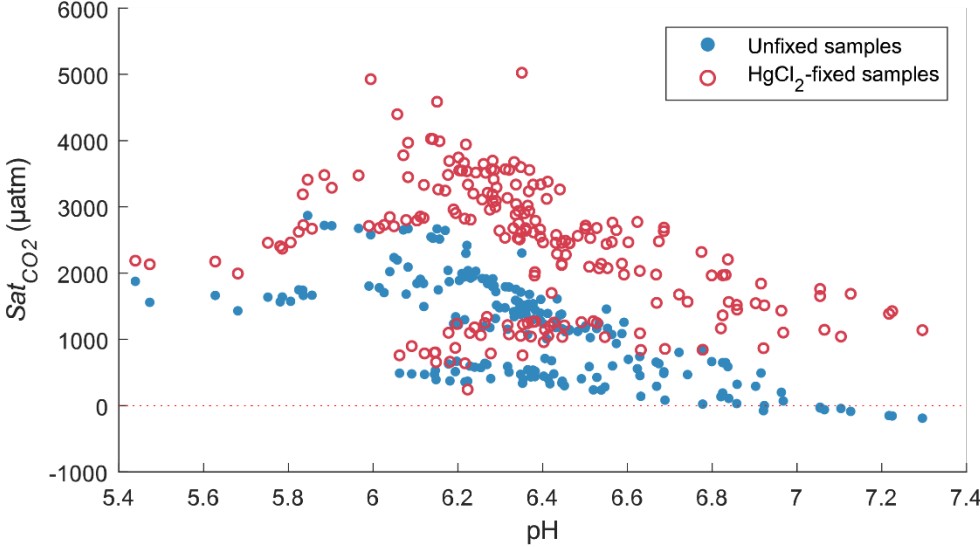

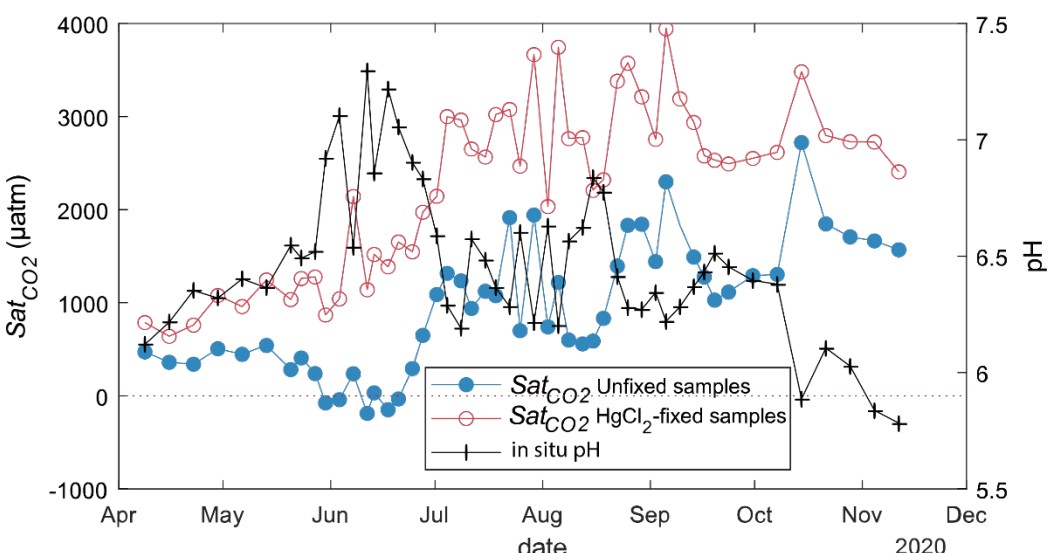

1027

**Fig. 4.** CO₂ saturation deficit ($Sat_{CO2}$) in Lake Lundebyvannet as a function of in situ pH for all

unfixed (obtained from DIC analysis) and HgCl₂-fixed (obtained from GC analysis) samples (top

panel). Timeseries of pH and CO₂ saturation deficit of surface water (1-m deep) for unfixed and

HgCl₂-fixed samples (bottom panel).

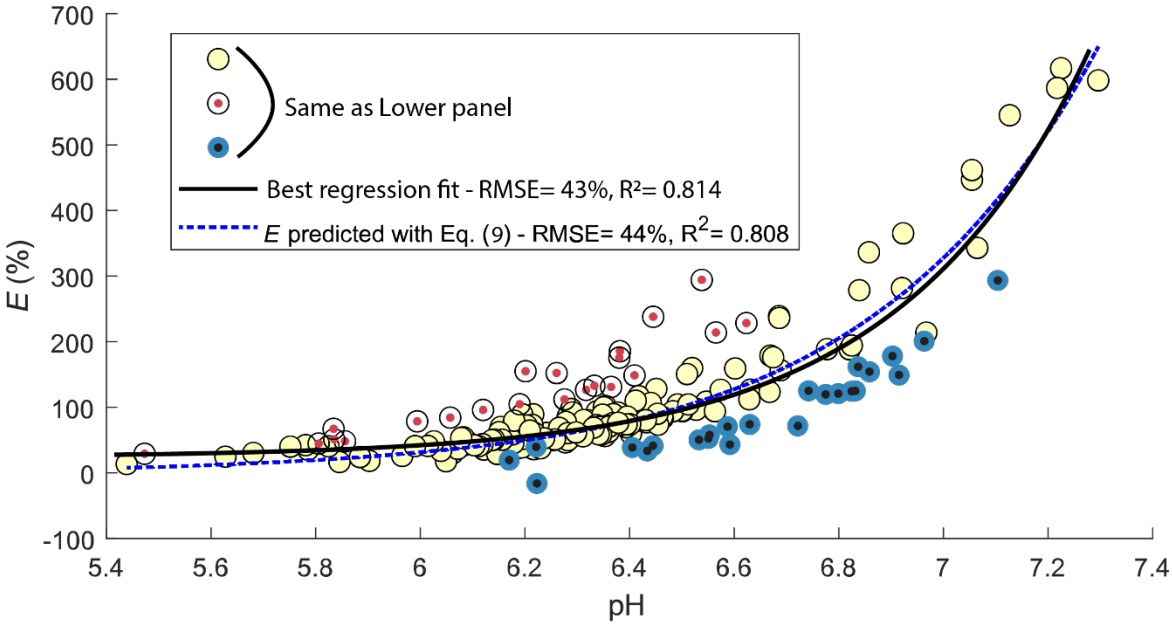

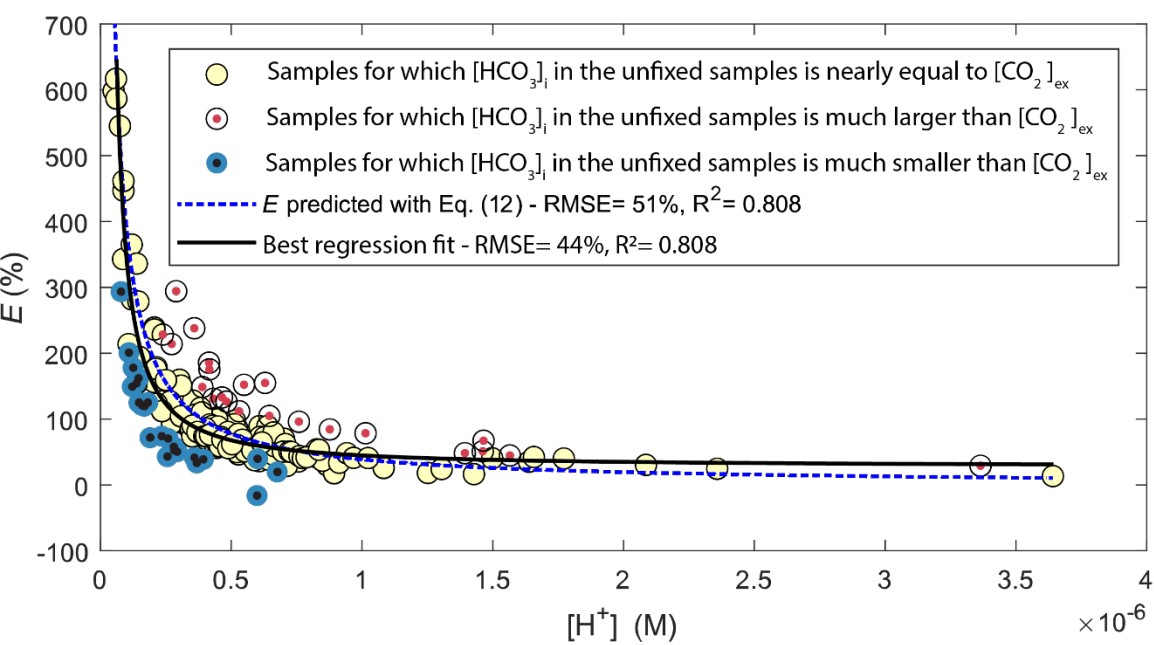

1032

**Fig. 5.** Comparison of observed (circles) and predicted (blue line) relative overestimation ($E$) of $CO_2$

concentrations caused by HgCl₂ fixation in Lake Lundebyvannet samples as a function of pH (top

panel) or proton concentration (bottom panel). The black line shows the best fit of the regression

analysis. White symbols represent samples for which the bicarbonate concentration in the unfixed

samples ($[HCO_3^-]_i$) is nearly equal to $CO_2$ overestimation ($[CO_2]_{ex}$), i.e., $\pm 20\mu M$ (equivalent to a pH

error of 0.05), while red and blue symbols represent samples for which initial bicarbonate

concentration was lower and higher than the $CO_2$ overestimation, respectively.

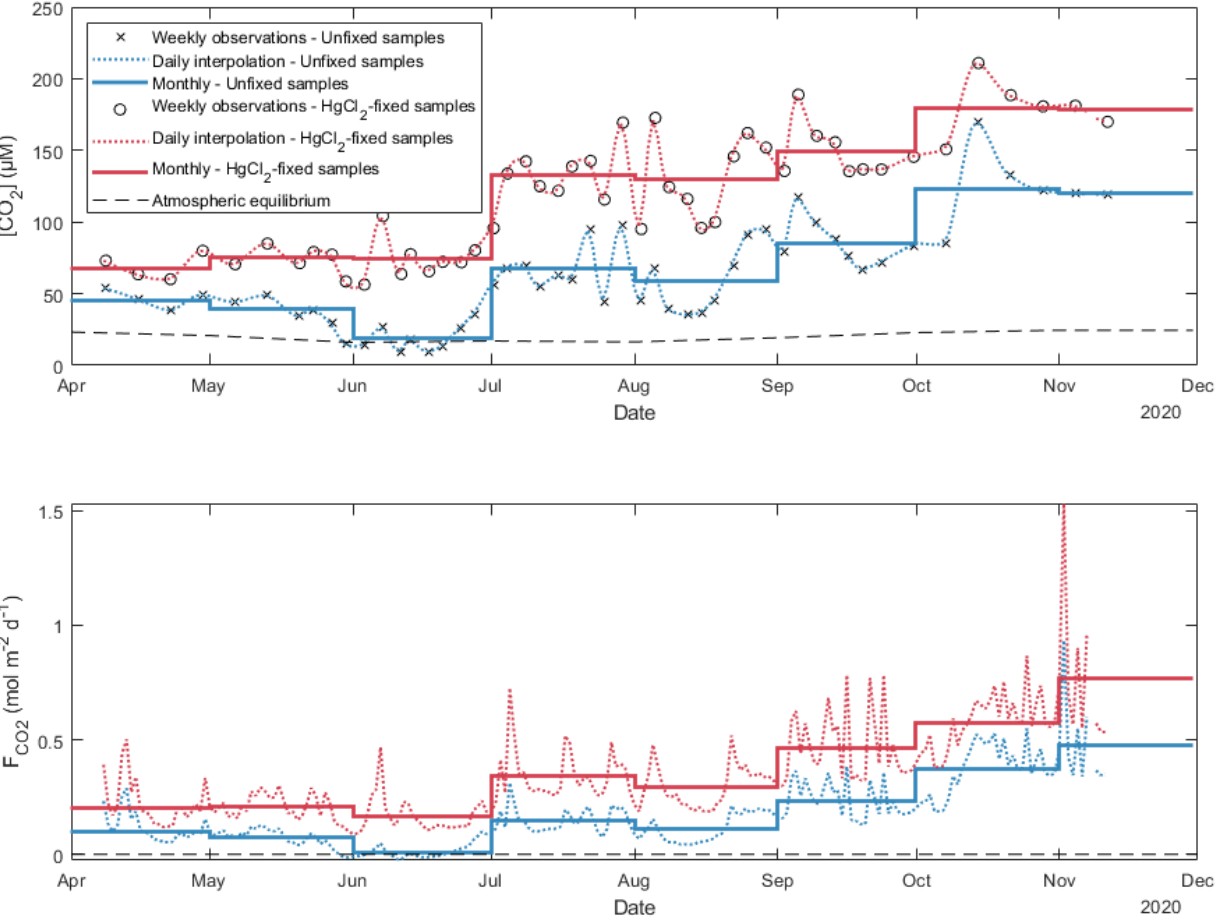


**Fig. 6.** Daily and monthly surface CO$_2$ concentrations ([CO$_2$]; top panel) and diffusion fluxes (F$_{CO2}$;
bottom panel) at the water-atmosphere interface from Lake Lundebyvannet (also in Tab. 3). Unfixed
samples were obtained by DIC analysis. Daily [CO$_2$] was interpolated from weekly data using a
modified spline (see text for details). Diffusion fluxes were calculated following Cole & Caraco

1045  (1998).

CO₂ overestimation from HgCl₂ fixation – Clayer et al.

**Tab. 1.** Stock and sample concentrations of $HgCl_2$, $CuCl_2$ and $AgNO_3$.

| Salt | Stock solution | Sample concentration | Rationale |
|---|---|---|---|
| $HgCl_2$ | 70 g/L (saturated) | 14.0±0.01 μg/mL (51.6 μM) | Dickson, Sabine & Christian, 2007 |
| $CuCl_2$ | 131.9 g/L | 26.4±0.02 μg/mL (154.7 μM) | $3 \times Hg$ |
| $AgNO_3$ | 87.6 g/L | 17.5±0.02 μg/mL (103.1 μM) | $2 \times Hg$ |

**Tab. 2.** pH and saturation indices of selected carbonate minerals estimated by PHREEQC for the unpreserved and preserved samples

| Preservatives | pH | Saturation indices | | | |
|---|---|---|---|---|---|
| | | $HgCO_3$ | $Cu_2(OH)_2CO_3$ - Malachite | $Cu_2(OH)_2CO_3$ - Azurite | $Ag_2CO_3$ |
| Unfixed | 6.72 | -2.31 | -4.96 | -8.71 | -16.42 |
| $HgCl_2$ | 6.40 | 3.64 | -5.89 | -10.10 | -17.20 |
| $CuCl_2$ | 6.45 | -2.55 | 2.26 | 2.11 | -17.44 |
| $AgNO_3$ | 6.71 | -2.31 | -4.97 | -8.73 | -4.33 |

**Tab. 3.** CO₂ concentrations ([CO₂], μM) and diffusion fluxes ($F_{CO2}$, mol m⁻² d⁻¹) from Lake Lake Lundebyvannet estimated from HgCl₂-fixed and unfixed samples following Cole and Caraco (1998). Ice-free season spans April to November. Data are also shown in Fig. 6.

| Preservatives | | Apr. | May | Jun. | Jul. | Aug. | Sep. | Oct. | Nov. | Ice-free season |
|---|---|---|---|---|---|---|---|---|---|---|
| [CO₂] | None | 45 | 39 | 19 | 68 | 59 | 85 | 123 | 120 | 67 |
| | $HgCl_2$ | 68 | 75 | 74 | 133 | 130 | 149 | 179 | 178 | 121 |
| | Diff (%) | +50 | +93 | +296 | +96 | +119 | +75 | +45 | +49 | +82 |
| $F_{CO2}$ | None | 0.10 | 0.07 | 0.01 | 0.15 | 0.11 | 0.23 | 0.37 | 0.48 | 0.17 |
| | $HgCl_2$ | 0.20 | 0.21 | 0.16 | 0.34 | 0.29 | 0.47 | 0.57 | 0.77 | 0.35 |
| | Diff (%) | +97 | +188 | +2163 | +130 | +162 | +99 | +55 | +62 | +108 |