# Peer review of "Technical Note: Preventing CO2 overestimation from mercuric or 1 copper (II) chloride preservation of dissolved greenhouse gases in 2 freshwater samples 3 4 5 François Clayer1\*, Jan Erik Thrane1, Kuria Ndungu1, Andrew King1, Peter Dörsch2<"

_EGUsphere, 2023_

## Referee Comment (RC1)

I am happy to see that these authors spent the time to do the analyses and writing of a technical note paper, as I believe it can prove very valuable for a wide audience and can improve the methods and results of many future studies. The use of certain chemicals to preserve samples for gas analysis is very common, and it is therefore really nice that there is research on the effects of the conservation by several methods. I therefore want to thank the authors of this manuscript for working on this topic, which I think is highly suitable for the Biogeosciences journal.

However, I am currently unable to assess the results of this paper, because I lack key information on the reliability of the results. I think this paper has a good potential, but the aspects that are missing are key for the interpretation of all results, so they need to be clarified before I could properly assess the quality of the whole paper.

I find it difficult that I cannot see if the actions of adding the substances to the samples itself changed something in the gas concentrations. There are differences in the gas concentrations at t0. This can be attributed to rapid effects of the interaction between the added chemicals and the sample, but it can also be due to sample contamination or degassing during handling. T0 is taken within 24 hours, but it remains unclear if this is after 1 hour for certain samples, and after 23 hours for others. There is also no miliQ or demineralized water control.
To have these issues addressed, in a reply to this comment but also in the manuscript, would give me more certainty about the results.

I would think it benefits the paper if results and discussion are separated into different sections, but I leave it up to the editor to decide on this, as I know there is also personal preference involved on that topic.

Another key issues that I would like to hear more on is why the t0 concentrations of $CO_2$ of the different treatments is that different. You write some about it, but as I see this as a major factor for many of your interpretations, I would like to have it addressed more in depth and more clearly. Why are there no error bars on the Cu and Hg boxes of $CO_2$?

Another important overarching issue is that it seems O2 significantly decreased in the samples with CuCl2. Is this indeed the case? This is of major importance, as it would suggest incomplete inhibition of microbial processes, and would affect all results for the CuCl2 treated samples.

Specific comments on certain parts of the manuscript are included below.

**Introduction**

The first paragraph is a bit odd. I think it is most important that the reader knows why it is needed to preserve the samples, not what they are used for in the end. I would put more emphasis on the processes in the bottles that will change the concentration.

I think it is important to also mention that ZnCl2 and CuCl2 are often used as toxic inhibitors. And give some details on why it seems certain researchers pick which inhibitor (or put that in the discussion).

Problems with the disposal of HgCl2 would be good to mention as well, as well as the costs associated with it.

48 + 49-50. Based on what did you decide to pick these papers as references?

75 – 81. The tone of this paragraph is also a bit odd. You state quite suddenly that $CO_2$ concentrations will be overestimated by HgCl2, while that is not really clear from the previous paragraphs. And the last line is way too firm for an introduction. The introduction should state the current state of the art, not opinions.

82 – 92. I would structure this paragraph differently. First, in neutral words (so no effective, overestimation, etc) explain what you investigated. Then state in a few sentence the key findings of your research.

**Methods**

96. Header 'study area' does not cover the content of the paragraph.
98. What is carefully collected? Explain in more detail.
100. Avoid = limit. And not gas loss, but gas exchange with the atmosphere.
107. That cannot be uniform for all Norwegian lakes, right? Do you mean that is the case for this specific lake, or do you mean this lake was like the lakes tested in that de Wit paper?
112. T=0 could be at any moment during the first 24 hours? Or was it the same for all samples? Were they randomized for the moment of analysis, or were some treatments done first and others later?
112. Were the same bottles measured at t0, t1 and t2, or were the bottles sacrificed?
113. 'as the water samples were collected'. Which water samples? Is it not the same as the 5L bottles? Unclear.
123. Unclear what 'it' means.
129. Why were they stored cold? In general, poisoned sampled are not kept cool, as far as I know.
133. Remove 'unfortunately'.
127. Was the same amount of liquid added to all samples? Was the miliQ flushed to remove the target gasses? What was the volume of the sample bottles?
141. Sealed with what?

**Results and discussion**

Fig. 1. Please split the graphs of O2 and $CO_2$ and CH4 and $N_2O$, it is now not clear that boxes 5-8 are a different gas.
100% saturation at which temperature? That of the lake water, at the 4 degrees of the storage, or the temperature during measurement?

Table 3. This table is a bit hard to read, while it does contain very interesting information. Can you add some lines or italic or bold text or something, to make it easier for the reader to focus?
What is the ice free season number made up of? Please explain in caption.
Also write in the caption what diff is (I know it seems obvious but still good to write down).

Fig. 3. Is the lower panel a useful addition? Or can I already get the same info from the upper panel?

Fig. 4. Isn't this the exact same info as in table 3? No need to show it twice. I think the graph is much nicer, it brings across your point very clearly.

357-362. Please address why the t0 concentration in the inhibited samples was so different from the control sample, and why the range in concentrations was much larger. Is it because of sample contamination with air during the addition of the inhibitors? If this is the case, that is not necessarily a bad thing, if you then also explain that that is one of the risks of inhibitor additons to samples.

382. Do you suggest that the microbial processes are not inhibited, or that there are abiotic proceses at play?
If it's the microbial processes, then how is it possible that these microbes are not inhibited, but the ones using O2 are?

386. Please mention whether there were statistically significant changes (between timepoints or between treatments) for N2.

408. In the CuCl2 treated samples, you have both O2 consumption and $CO_2$ production. Why do you think these are not linked?

I have not provided detailed comments on the later sections, as I think it important to know more about the $CO_2$ results first, like I stated in my starting comments.

---

## Author Comment (AC1)

We are grateful to the reviewer, Thank you so much for this thorough, constructive, and in-depth review. Thank you for pointing out concrete needs for clarifications and improvements. Below we provide some preliminary response to the main comments and all numbered comments. We will also provide detailed response to technical corrections at a later stage. (Reviewer's comments are underlined for clarity).

Specific comments:

- Are the studied lakes representative for other lakes or waterbodies? Please clearly outline limitations of this study in terms of impact and application in a broader sense.

Thank you for raising this. Admittedly, the significance of our study and representativeness was somewhat neglected. Addressing this will definitely improve the manuscript. Short answer is "Yes", Svartkulp is particularly representative of Northern Hemisphere lakes, typically found in granitic bedrock regions in North-East America and Scandinavia. It is a typical low-productivity, heterotrophic, slightly acidic to neutral, moderately humic lake. Similar lakes are found in Southern Norway (de Wit et al., 2023), large parts of Sweden (Valina et al. 2014), and Finland, Atlantic Canada (Houle et al., 2022), Ontario, Québec and North-East USA (Skjelkvåle and de Wit 2011; Weyhenmeyer et al., 2019). Note that most of the lakes in Norway are more acidic than Svartkulp (de Wit et al., 2023), and have lower alkalinity and would therefore be more sensitive to acidification caused by HgCl2 fixation. Even if Svartkulp is among the best buffered lakes in Norway, our findings are also relevant to more acidic, lower ionic strength lakes found in Norway and large parts of Northern Canada.

Lundebyvannet, is also representative of a large group of these Noorthern lakes, however, it is quite a productive lake with high photosynthetic activity, which is more of a end-member case (e.g., worst-case scenario for Norway, related to CO2 flux overestimation with HgCl2 fixation).

- Is it feasible to assume unfixed samples to represent "real" concentrations/fluxes, as control? Could you discuss this further and eventually consider renaming "control" to "unfixed" for the first experiment?

You agree with the suggestion to rename "control" samples to "unfixed" samples. In fact, we should reorganize the discussion and start with a discussion on how representatives these unfixed samples are to real conditions.

For CO2 and O2, processing unfixed samples kept dark and cold is considered "real concentrations". In particular, a few studies have used unfixed samples (only preserved dark at 4C) up to 48h after sampling to determine CO2 or DIC concentrations (e.g., Sobek et al. 2003, Kokic et al., 2015). So, the DIC samples in Lundebyvannet and the CO2 and O2 concentrations at t=0 should be fine. We will refer to these studies in the revised manuscript.

However, for CH4, Fig. 1 shows that already within 24h, the CH4 concentration in the unfixed sample is below atmospheric saturation while it is consistently much higher in all three sets of fixed samples. So, unfixed samples do not represent real CH4 concentrations.

- The Methods section lacks necessary detail. I would suggest to restructure the section to make the experimental setup and respective study lakes clearer. E.g. in the study area section Lake Lundebyvannet should also be introduced, and ideally both lakes should be presented with the same level of detail relevant to the respective experiments. More importantly, I'm missing information on sampling procedures and their feasibility. Since this

is a technical note, I believe the Methods should be sound. I added several comments in this regard below.

Excellent point, yes we will add all necessary details, better describe Lundebyvannet. It is right that the sampling procedure needs to be clearer for others to be able to reproduce it and to evaluate their feasibility in other field contexts. Thank you for your help and all the specific line comments related to this, this is very helpful.

- I think the results of this study could be put into clear recommendations for future studies, and this could be part of the abstract and expressed more clearly in the discussion.

Another nice suggestion. We will add a section on recommendations in the Discussion and represent this section in the abstract. While some recommendations are formulated L. 535 onwards, they deserve a stronger position in the manuscript.

- The Introduction could benefit from adding some information about the other preservatives studied. The application of HgCl2 is broadly introduced (could be shortened), but the description of the substitutes dealt with in this study falls short. Are there any other studies where CuCl2 or AgNO3 were used to determine dissolved gas concentrations? Are there differences expected between the application of those two?

We had a hard time trying to find studies where CuCl2 and AgNO3 have been used as preservatives for freshwater samples for the determination of dissolved gases. In fact, these preservatives are often only briefly mentioned in the methods which makes it difficult to identify relevant studies. We can nevertheless re-equilibrate the introduction between HgCl2 and the two other alternatives by describing the expected impacts on water chemistry of the fixation with AgNO2 and CuCl2.

Please note, the numbers at the beginning of each comment denote the line numbers.

1. 26-27: can you be more specific about time periods (3w, 3m)?

Yes, initial increase to from 24h to 3 weeks, and then decrease from 3 weeks to 3 months.

2. 29: are low ionic strength / high DOC lakes representative?

Yes, as describe above, we will rephrase this (see our response to your first comment above)

3. 30: are these estimations valid for other lakes?

These are probably among the worst case scenarios regarding over-estimation of CO2 efflux, we will clarify this in the abstract and discussion.

4. 31: I think explicitly adding recommendations here would be useful.

Yes, will do (see also our response to your 4th comment above)

5. 59: better in regards to what?

"yielded concentrations after preservation and storage closer to real concentrations", will clarify this.

6. 67: what is the impact of higher H+ concentrations?

Shift in the equilibrium between CO2 and HCO3 with consequences on CO2 and DIC estimations.

7. 70: could you elaborate why this leads to an overestimation of CO2 concentration?

Yes, we will elaborate here. If the pH is lower than thought, the measured CO2 signal by the GC represents a higher proportion of DIC, compared with a situation where the pH equals the expected pH.

8. 82-92: it would help the reader if it was made clearer here that two different experiments were conducted and two different lakes were sampled for that, for example by 1)... 2)...

As suggested, we will introduce the distinction here.

9. 87: This assumes that unfixed samples are the control, or "real results". Is that feasible?

We will rephrase here also consistent with our response to your second comment above.

10. 99: Did you collect the water from the surface? Did you use anything other than the bottles to avoid bubbling or degassing? Were samples temperature controlled (or otherwise controlled) between sampling and analysis?

We will describe this better.

11. 100: slowly poured - a bit vague? How could you guarantee no degassing?

We will describe in details the procedure, e.g., "The sample bottles were almost horizontal when starting to pour and slowly tilted as the water level was increasing to avoid any turbulence, bubbling and the like".

12. 103: Are the results you got from the water samples representative for lakes in the region in terms of magnitude? Do the numbers represent means of the sub-samples or was each sub-sample used for determination of one of the parameters?

Yes, see also our response to your 1st comment.

13. 105: As far as I understand, the concentration of platinum does not necessarily describe the color characteristics of water?

We will come back to this, I believe these are standard units, but color is not a major parameter here, DOC, Alkalinity, and nutrients are more important.

14. 106: how did you measure the temperature? is this an important information if the water was transported to the lab? Or did you preserve this temperature during transport?

This will be clarified, 18.5C was in the lake, we also monitored the temperature in the lab.

15. 111: technically 3 treatments and one control. Which of the scenarios would presumably result in the most "real" concentration?

This depends on which gas, CH4 is closer to real concentrations in the fixed samples, while CO2 and O2 in the unfixed samples at t=0 is the closer to real concentrations as also described in our response to your 2nd comment.

16. 112: why did you choose these time steps? could you elaborate if these times are representative?

These timesteps were selected and thought to be representative for average storage times, 24h as "direct measurements following a field trip", and 3 months as a long term storage scenario.

17. 116: Is the preparation of the solutions part of the experiment? do the yielded concentrations have an uncertainty? or would a derivation not have an impact on the outcome?

We used standard chemistry lab equipment with high accuracy scale to weigh the salt and volumetric flasks. Reference to the suppliers of the preservative salts will also be included in the description.

18. 133: Did the fact that pH was not measured affect your study? Or was that one reason to use the PHREEQC model?

Admittedly, the experimental design was imperfect and pH measurements following fixation should have been included. It would have yielded valuable information on the impact of preservation addition on e.g., pH. So yes, part of the reason why we used the PHREEQC model was because we didn't have pH data. However the PHREEQC model proved to be much more useful than just estimating the pH upon preservative addition, it was also predicting if the precipitation of some mineral phases was plausible.

19. 137: Is the sampling strategy outlined different than the one for the first experiment? How did you achieve sampling water from different depths? It would also be nice if both lakes were described with the same level of detail.

Agreed, both lakes need to be described with the same level of details. This will be corrected. Water was collected from different depths using a standard water sampler (information on this will be included). The sampling procedures will be described in more details here. They included direct sampling on the lake in this case.

20. 145: Could you clarify the purpose of DIC analysis in this study?

DIC analyses were performed to obtain an independent estimation of CO2 and DIC concentration, compared to the GC analysis. This will be added here.

21. 147: Add name of TOC analyzer and/or merge with other sections below to avoid repetition. You state that samples were not fixed – why not?

Samples were not fixed to avoid any interaction with the sample. However, the samples were analyzed within 2h. Note that this is in line with previous studies.

22. 151: Did you compare the pH data with that measured with the pH-meter as mentioned above, or why measure twice?

We apologize, this is an error. pH was not measured in the laboratory, we only used the pH data from the *in situ* HOBO sensor. This sentence will be corrected.

23. 160: The temperature was recorded during shaking – do you mean the water temperature? What was the purpose?

Yes, the water temperature (sorry for the lack of clarity) was recorded during shaking to ensure it is stable throughout the equilibrium process. The temperature was stable, and the value was used to calculate back the dissolved gas concentrations.

24. 171: Do you mean ambient air was used for calibration? Did you know the concentrations of the ambient air?

Yes, ambient air was used for O2 and N2 calibration. Ambient air is measured regularly and is stable through time.

25. 175 section: I think it would help to directly add formulas in a section in the appendix for better understanding and reproducibility.

Good suggestion, thank you.

26. 187: Could you explain the purpose of DIC analysis here or in earlier sections. Are the CO2 concentrations calculated in addition to the concentrations measured by GC for comparison? I think it may not always be clear where you used measured or calculated CO2 concentrations.

This will be introduced earlier, see comment #20-

27. 244: Is it important to mention what files were input and output files? For someone who doesn't know the program, this info seems meaningless.

We will clarify this.

28. 255: Is this analysis done in retrospect to make up for not measuring sample pH directly after storage (among other things)?

Yes partly, this will be clarified.

29. 320: What temperature did you use to determine the Schmidt number?

I believe we used the ambient water temperature, but I need to confirm this and add this information here, agreed.

30. 328 f: It is nice to have different temporal resolutions, but what is the purpose of that for this study? Would your measurements not reflect instantaneous fluxes (could maybe be considered as daily fluxes) rather than weekly?

The main idea to show fluxes with these three temporal aggregations is to highlight the magnitude of the mis-estimation of the fluxes when HgCl2 is used as preservatives for the water samples. The error magnitude can be much larger over shorter timescales (see. E.g., Fig. 4).

31. 340: A rather general comment to this study: what is the assumption about the development of gas concentration in between times 0, 3w, 3m? E.g., what would the concentration after 2w or 2m supposedly look like? Did you examine that?

We haven't looked at other time points, this is difficult to predict.

32. 362: Would preservation with AgNO3 then be preferable rather than with HgCl2 due to its toxicity? Can you draw conclusions regarding CH4 from your results?

Yes definitely. These recommendations will be summarized into a dedicated section in the discussion later on.

33. 364, Fig. 1: Concentrations of all gases (except CO2) show largest ranges for AgNO3 addition (largest bars) after 3w. Is there an explanation for that? Add to caption: what do the boxplots show, presumably 25th and 75th percentiles and the median?

Unfortunately, no we have no explanation for the relatively larger range for the AgNO3 fixed samples after 3w. We will add the description of the boxplots in the caption, yes.

34. 375 f: Are you arguing that this process is slowed down in freshwater?

Not necessarily slowed down because of freshwaters, there are many parameters playing a potential role here, e.g., temperature, substrate concentration, etc. Rees et al. (2021) performed their incubations at ambient temperatures which is the most likely explanation for the difference seen with our observations. Our samples were stored at 4C for 3 months.

35. 380-381: Is this assumption reflected in your results by the decrease of N2 concentration? Is there also an explanation for the N2O consumption following production?

The interpretation of small changes in the N2 data should be avoided since none of the groups are significantly different from each other.

36. 385: Did you perform a statistical test here too? Is it worthwhile mentioning that the concentrations seem to have the opposite response over time than N2O?

These changes do not have the same magnitude, µM for N2, nM for N2O. The expected changes in N2 from the process mentioned in comment #35 is not detectable.

37. 422: The opposite of what you state in the text is shown in Tab. 3. Is there a mistake in the labels?

Yes, there is a mistake in the labels in Table 3, these are not consistent with Fig. 4. Thank you for spotting that.

38. 426 f: Wouldn't we expect to see a shift in pH then in Fig. 2 (top panel)? It appears as if the fixed and unfixed samples have the same pH?

The pH plotted here in the *in situ* pH which is the same for both sample sets. pH was determined in the lab only on DIC samples, not on GC samples. This will be clarified in the caption.

39. 435, Tab. 3: What was the reason to show fluxes calculated following Cole and Caraco and not the other wind-based models here?

This was to avoid overloading the table, the values are different but the relative differences between fixed and unfixed samples is bound to the original concentration data. We believe there is no point in showing three models showing the same differences.

40. 466: Why was this cut-off of 20 µM chosen?

This 20 µM cut-off was chosen as the maximum likely error from e.g., pH error of 0.05. This will be clarified in the caption.

41. 503: Which of those shown in Tab. 3 and Fig. 4 were obtained from DIC analyses?

We apologize, there is an inversion in the "Preservatives" labels in Table 3, lower values (1st and 4th rows) are from "unfixed" samples analyzed for DIC while the highest values were those from samples fixed with HgCl2 (2nd and 5th rows). Same in Fig. 4, blue lines and crosses (lower values) are from "unfixed" samples analyzed for DIC.

42. 507 f: This estimate is only valid for the tested lakes. What would be the implication for other lakes? Is this also valid for sea water samples? Do you have recommendations or a protocol that should be followed? And what about greenhouse gases other than CO2?

Excellent questions, thank you. We will answer those questions in a specific section in the discussion. The overestimation of CO2 concentration of up to 300% is likely among the worst-case scenarios since it is restricted to lakes which present both a slightly acidic to near neutral pH and intensive photosynthetic activity. However, such lakes with similar pH and productivity are relatively abundant in the Northern Hemisphere (see our response to the first comment). Note that this overestimation is not valid for sea water samples where the fixation with HgCl2 only caused a unsignificant acidification because of the high buffering capacity of seawater (see Chou et al. 2016). For the other greenhouse gases, AgNO3 seems to be as good as, or slightly better than HgCl2 and CuCl2 for preservation. All of these findings will be further discussed and presented as recommendations in a specific section in the discussion.

**References:**

Chou W.C., Gong G.C., Yang C.Y. & Chuang K.Y. (2016) A comparison between field and laboratory pH measurements for seawater on the East China Sea shelf. Limnology and Oceanography-Methods, 14, 315-322. https://doi.org/10.1002/lom3.10091

Clayer, F., Thrane, J.-E., Brandt, U., Dörsch, P., & de Wit, H. A. (2021). Boreal Headwater Catchment as Hot Spot of Carbon Processing From Headwater to Fjord. Journal of Geophysical Research: Biogeosciences, 126(12), e2021JG006359. https://doi.org/10.1029/2021JG006359

de Wit, H. A., Garmo, Ø. A., Jackson-Blake, L. A., Clayer, F., Vogt, R. D., Austnes, K., Kaste, Ø., Gundersen, C. B., Guerrerro, J. L., & Hindar, A. (2023). Changing Water Chemistry in One Thousand Norwegian Lakes During Three Decades of Cleaner Air and Climate Change. Global Biogeochemical Cycles, 37(2), e2022GB007509. https://doi.org/10.1029/2022GB007509

Houle, D., Augustin, F., & Couture, S. (2022). Rapid improvement of lake acid–base status in Atlantic Canada following steep decline in precipitation acidity. Canadian Journal of Fisheries and Aquatic Sciences, 79(12), 2126–2137. https://doi.org/10.1139/cjfas-2021-0349

Kokic, J., Wallin, M. B., Chmiel, H. E., Denfeld, B. A., & Sobek, S. (2015). Carbon dioxide evasion from headwater systems strongly contributes to the total export of carbon from a small boreal lake catchment. Journal of Geophysical Research: Biogeosciences, 120(1), 13–28. https://doi.org/10.1002/2014JG002706

Rees, A. P., Brown, I. J., Jayakumar, A., Lessin, G., Somerfield, P. J., & Ward, B. B. (2021). Biological nitrous oxide consumption in oxygenated waters of the high latitude Atlantic Ocean. Communications Earth & Environment, 2(1), Article 1. https://doi.org/10.1038/s43247-021-00104-y

Skjelkvåle, B. L., & de Wit, H. A. (2011). Trends in precipitation chemistry, surface water chemistry and aquatic biota in acidified areas in Europe and North America from 1990 to 2008 (ICP Waters report 106/2011). In 126. Norsk institutt for vannforskning. https://niva.brage.unit.no/niva-xmlui/handle/11250/215591

Sobek, S., Algesten, G., Bergström, A.-K., Jansson, M., & Tranvik, L. J. (2003). The catchment and climate regulation of pCO2 in boreal lakes. Global Change Biology, 9(4), 630–641. https://doi.org/10.1046/j.1365-2486.2003.00619.x

Valiente, N., Eiler, A., Allesson, L., Andersen, T., Clayer, F., Crapart, C., Dörsch, P., Fontaine, L., Heuschele, J., Vogt, R., Wei, J., de Wit, H. A., & Hessen, D. O. (2022). Catchment properties as predictors of greenhouse gas concentrations across a gradient of boreal lakes. 10(880619). https://doi.org/10.3389/fenvs.2022.880619

Valinia, S., Englund, G., Moldan, F., Futter, M. N., Köhler, S. J., Bishop, K., & Fölster, J. (2014). Assessing anthropogenic impact on boreal lakes with historical fish species distribution data and hydrogeochemical modeling. Global Change Biology, 20(9), 2752–2764. https://doi.org/10.1111/gcb.12527

Weyhenmeyer, G. A., Hartmann, J., Hessen, D. O., Kopáček, J., Hejzlar, J., Jacquet, S., Hamilton, S. K., Verburg, P., Leach, T. H., Schmid, M., Flaim, G., Nõges, T., Nõges, P., Wentzky, V. C., Rogora, M., Rusak, J. A., Kosten, S., Paterson, A. M., Teubner, K., … Zechmeister, T. (2019). Widespread diminishing anthropogenic effects on calcium in freshwaters. Scientific Reports, 9(1), Article 1. https://doi.org/10.1038/s41598-019-46838-w

---

## Author Comment (AC2)

We are grateful to the reviewer, Thank you so much for this constructive and detailed review. Below we provide some preliminary response to the each reviewr's comments. (Reviewer's comments are underlined for clarity).

Specific comments:

- I am happy to see that these authors spent the time to do the analyses and writing of a technical note paper, as I believe it can prove very valuable for a wide audience and can improve the methods and results of many future studies. The use of certain chemicals to preserve samples for gas analysis is very common, and it is therefore really nice that there is research on the effects of the conservation by several methods. I therefore want to thank the authors of this manuscript for working on this topic, which I think is highly suitable for the Biogeosciences journal.

Thank you, it is nice to see that our work was appreciated.

- However, I am currently unable to assess the results of this paper, because I lack key information on the reliability of the results. I think this paper has a good potential, but the aspects that are missing are key for the interpretation of all results, so they need to be clarified before I could properly assess the quality of the whole paper.

We are sorry to read this and apologize for taking your time, we will try to improve the manuscript and clarify all unclear points raised below.

- I find it difficult that I cannot see if the actions of adding the substances to the samples itself changed something in the gas concentrations. There are differences in the gas concentrations at t0. This can be attributed to rapid effects of the interaction between the added chemicals and the sample, but it can also be due to sample contamination or degassing during handling. T0 is taken within 24 hours, but it remains unclear if this is after 1 hour for certain samples, and after 23 hours for others. There is also no miliQ or demineralized water control. To have these issues addressed, in a reply to this comment but also in the manuscript, would give me more certainty about the results.

Unfortunately, the experimental design was imperfect, and would have been nice to have some benchmark steps such as pH measurements of the samples right after preservative addition. However, we can easily rule out sample contamination and degassing during handling, we have a really high reproducibility between the 6 replicates, and the relative difference between the unfixed and fixed samples are not consistent with degassing, this will be described in the results section.

Regarding the time for T=0 samples, our wording was too vague, we apologize for that. All samples were treated at the same time within 1h or 2h (this includes transfer from the 5L bottles to the 120mL glass bottle and fixation), we will dig up the exact timing, kept dark and cold overnight and then analyzed on the GC the next day.

If this is required, we could run a simple additional field test with lake water, adding the preservatives, and measuring the pH of the sub-samples 24h after fixation (and include a demineralized water control). Although this would not be the same sample batch (the water might have a slightly different composition), we would follow the same sampling and handling procedures.

- I would think it benefits the paper if results and discussion are separated into different sections, but I leave it up to the editor to decide on this, as I know there is also personal preference involved on that topic.

I believe, we agree on this one and also think the manuscript would be clearer with results first.

- Another key issues that I would like to hear more on is why the t0 concentrations of CO2 of the different treatments is that different. You write some about it, but as I see this as a major factor for many of your interpretations, I would like to have it addressed more in depth and more clearly. Why are there no error bars on the Cu and Hg boxes of CO2?

The large difference on CO2 concentrations at T=0 is due to rapid interaction with the preservatives. Maybe this additional field test would also help document this rapid acidification of the samples with HgCl2 and CuCl2. The description of such a test would be included in the supplementary material.

- Another important overarching issue is that it seems O2 significantly decreased in the samples with CuCl2. Is this indeed the case? This is of major importance, as it would suggest incomplete inhibition of microbial processes, and would affect all results for the CuCl2 treated samples.

Yes, there is a significant decrease in O2 over time which is likely due to incomplete inhibition of microbial processes. We didn't focus on this earlier because the acidification of the samples was already a sufficient reason for disapproving its use as preservative. We will also describe this effect in a clearer manner and warn about potential incomplete inhibition with CuCl2.

- Specific comments on certain parts of the manuscript are included below.

**Introduction**

- The first paragraph is a bit odd. I think it is most important that the reader knows why it is needed to preserve the samples, not what they are used for in the end. I would put more emphasis on the processes in the bottles that will change the concentration.

Agreed, and thank you, we can easily streamline this paragraph.

- I think it is important to also mention that ZnCl2 and CuCl2 are often used as toxic inhibitors. And give some details on why it seems certain researchers pick which inhibitor (or put that in the discussion). Problems with the disposal of HgCl2 would be good to mention as well, as well as the costs associated with it.

Thank you, we will do so and mention the problem about HgCl2 disposal.

- 48 + 49-50. Based on what did you decide to pick these papers as references?

We performed a wide literature search for studies specifically mentioning HgCl2 as preservative for dissolved gas samples in their methods.

- 75 – 81. The tone of this paragraph is also a bit odd. You state quite suddenly that CO2 concentrations will be overestimated by HgCl2, while that is not really clear from the previous paragraphs. And the last line is way too firm for an introduction. The introduction should state the current state of the art, not opinions.

We will adapt the end of the previous paragraph and adjust the tone here. Thank you for this suggestion.

- 82 – 92. I would structure this paragraph differently. First, in neutral words (so no effective, overestimation, etc) explain what you investigated. Then state in a few sentence the key findings of your research.

Agreed, and

**Methods**

- 96. Header 'study area' does not cover the content of the paragraph.

Agreed, the study sites will be introduced first and then "Sampling procedure" will be used as the section title.

- 98. What is carefully collected? Explain in more detail.

The sample procedure will be better described here.

- 100. Avoid = limit. And not gas loss, but gas exchange with the atmosphere.

Agreed.

- 107. That cannot be uniform for all Norwegian lakes, right? Do you mean that is the case for this specific lake, or do you mean this lake was like the lakes tested in that de Wit paper?

This lake is like the set of 1000 lakes, carefully selected to be representative for Norwegian lakes, presented in de Wit et al. (2023).

- 112. T=0 could be at any moment during the first 24 hours? Or was it the same for all samples? Were they randomized for the moment of analysis, or were some treatments done first and others later?

T=0 samples (as the other time points) were all treated at the same time, and also randomized, yes. This will be clearly described.

- 112. Were the same bottles measured at t0, t1 and t2, or were the bottles sacrificed?

The bottles were sacrificed to avoid introducing a headspace and sample dilution etc. This is why we had 72 bottles (6 replicates x 3 time-points x 4 treatments). This will be clarified.

- 113. 'as the water samples were collected'. Which water samples? Is it not the same as the 5L bottles? Unclear.

Yes, but the 5L bottle was filled in the field, and up return in the lab, the 72 120mL bottles were filled.

- 123. Unclear what 'it' means.

"it" refers to "estimated toxicity". This will be rephrased.

- 129. Why were they stored cold? In general, poisoned sampled are not kept cool, as far as I know.

This is how we have worked so far to be on the safe side (see e.g., Clayer et al. 2021), and this also prevents the samples to be subjected to changes in room temperature. Keeping water samples cold

and dark is the best way to optimize sample preservation. There is no guarantee that poisoned samples will be 100% inert, keep them cold helps to limit any microbial activity.

- 133. Remove 'unfortunately'.

Will do.

- 127. Was the same amount of liquid added to all samples? Was the miliQ flushed to remove the target gasses? What was the volume of the sample bottles?

Yes, the same volume of 240μL was added to all bottles. And no miliQ water was not flushed, but given its small volume, its concentration is unsignificant for the water samples. This will be clarified.

- 141. Sealed with what?

The samples were sealed with gas tight butyl rubber stoppers as for the first experiment. We used the sample bottles and caps for both experiments. This will be clarified.

**Results and discussion**

- Fig. 1. Please split the graphs of O2 and CO2 and CH4 and N2O, it is now not clear that boxes 5-8 are a different gas. 100% saturation at which temperature? That of the lake water, at the 4 degrees of the storage, or the temperature during measurement?

Everything is reported back to *in situ* temperature. This will be clarified, sorry for this oversight. In fact, saturation is only relevant for *in situ* conditions since after sampling, the water samples were never in contact with atmosphere. Thank you for the nice suggestions, we will split the graphs here and clarify the caption.

- Table 3. This table is a bit hard to read, while it does contain very interesting information. Can you add some lines or italic or bold text or something, to make it easier for the reader to focus? What is the ice free season number made up of? Please explain in caption. Also write in the caption what diff is (I know it seems obvious but still good to write down).

The style of the journal is to avoid additional line, but we can remove the "%" symbol on row 3 and 6, remove the decimal on the row 1 and 2, and arrange the alignments (probably increase the spacing a little) to improve visibility. The ice-free season is the average over the whole experiment duration from April to November. This will be clarified in the caption.

Note also that the "Preservatives" labels were inversed, lower values (1st and 4th rows) are from "unfixed" samples analyzed for DIC while the highest values were those from samples fixed with HgCl2 (2nd and 5th rows).

- Fig. 3. Is the lower panel a useful addition? Or can I already get the same info from the upper panel?

- Fig. 4. Isn't this the exact same info as in table 3? No need to show it twice. I think the graph is much nicer, it brings across your point very clearly.

Table 3 and Fig. 4 both display the monthly mean, yes. In addition, Fig. 4 displays the observations and interpolated daily data. Table 4 also shows the relative difference between fixed and unfixed

samples in % as well as the mean for the whole duration of the experiment. Since this is a technical paper, we believe it is useful to keep both for clarity and depending on the reader's affinity for table or figure. However, we agree that we should specifically state that these two objects display the same data. This will be clarified in the captions.

- 357-362. Please address why the t0 concentration in the inhibited samples was so different from the control sample, and why the range in concentrations was much larger. Is it because of sample contamination with air during the addition of the inhibitors? If this is the case, that is not necessarily a bad thing, if you then also explain that that is one of the risks of inhibitor additons to samples.

The $CH_4$ concentration at T=0 in the unfixed sample is below atmospheric saturation, as shown by the horizontal line in Fig. 1. In contrast, we expect the $CH_4$ concentration to be oversaturated in "real conditions", as supported by Valiente et al. 2022, Clayer et al. 2021. The $CH_4$ concentration at t=0 in the fixed samples are consistently between 0.2 to 0.33 µM, independently of the preservative which suggests that $CH_4$ in the unfixed samples has been consumed (which is realistic considering oxic methanotrophy reaction rates, see L. 369-371) while the concentration in the fixed samples is close to real conditions.

The reason why the range of $CH_4$ concentrations in the fixed samples is larger is likely due to minor gas losses during handling. In fact, at the lab temperature (which is higher than *in situ* lake temperature) and after some storage time, the water samples will have a higher total gas oversaturation than at the start of the experiment. Under these conditions, $CH_4$, being the least soluble of the gases, will be lost first. Such $CH_4$ losses are much smaller for the "control unfixed" samples which show much lower $CH_4$ concentrations. This will be clarified in the manuscript.

- 382. Do you suggest that the microbial processes are not inhibited, or that there are abiotic proceses at play? If it's the microbial processes, then how is it possible that these microbes are not inhibited, but the ones using O2 are?

Yes, we suggest that microbial processes are not completely inhibited. We will make this point clearer. Note that certain metabolic pathways can be selectively inhibited and $N_2O$ is likely much more sensitive (being at 10's nM levels) to subtle microbial activity than $O_2$ (being at 100's µM levels).

- 386. Please mention whether there were statistically significant changes (between timepoints or between treatments) for N2.

There were no significant changes between timepoints or between treatments for $N_2$. This will be added.

- 408. In the CuCl2 treated samples, you have both O2 consumption and CO2 production. Why do you think these are not linked?

Yes, good point. $CO_2$ production in the $CuCl_2$ treated samples is likely partially link to $O_2$ consumption through microbial respiration. However, the $CO_2$ production being much larger than $O_2$ consumption, an additional source of $CO_2$ is needed.

- I have not provided detailed comments on the later sections, as I think it important to know more about the CO2 results first, like I stated in my starting comments.

We hope, we have provided some clarifications and will continue to do so at a later stage if the manuscript is going through the next stage. Thank you for your assessment.

References:

Clayer, F., Thrane, J.-E., Brandt, U., Dörsch, P., & de Wit, H. A. (2021). Boreal Headwater Catchment as Hot Spot of Carbon Processing From Headwater to Fjord. Journal of Geophysical Research: Biogeosciences, 126(12), e2021JG006359. https://doi.org/10.1029/2021JG006359

de Wit, H. A., Garmo, Ø. A., Jackson-Blake, L. A., Clayer, F., Vogt, R. D., Austnes, K., Kaste, Ø., Gundersen, C. B., Guerrerro, J. L., & Hindar, A. (2023). Changing Water Chemistry in One Thousand Norwegian Lakes During Three Decades of Cleaner Air and Climate Change. Global Biogeochemical Cycles, 37(2), e2022GB007509. https://doi.org/10.1029/2022GB007509

Valiente, N., Eiler, A., Allesson, L., Andersen, T., Clayer, F., Crapart, C., Dörsch, P., Fontaine, L., Heuschele, J., Vogt, R., Wei, J., de Wit, H. A., & Hessen, D. O. (2022). Catchment properties as predictors of greenhouse gas concentrations across a gradient of boreal lakes. 10(880619). https://doi.org/10.3389/fenvs.2022.880619

---

## Author Response (AR1)

Title: Technical Note: Preventing CO2 overestimation from mercuric or copper (II) chloride preservation of dissolved greenhouse gases in freshwater samples

Reviewers' comments are underlined for clarity. Line numbers refer to line in the revised (clean version) version of the manuscript, if not specified otherwise.

**REVIEWER REPORT(S):**

**Reviewer 1:**

We are grateful to the reviewer, Thank you so much for this constructive and detailed review. Below we provide some preliminary response to the each reviewer's comments.

Specific comments:

- I am happy to see that these authors spent the time to do the analyses and writing of a technical note paper, as I believe it can prove very valuable for a wide audience and can improve the methods and results of many future studies. The use of certain chemicals to preserve samples for gas analysis is very common, and it is therefore really nice that there is research on the effects of the conservation by several methods. I therefore want to thank the authors of this manuscript for working on this topic, which I think is highly suitable for the Biogeosciences journal.

Thank you, it is nice to see that our work was appreciated.

- However, I am currently unable to assess the results of this paper, because I lack key information on the reliability of the results. I think this paper has a good potential, but the aspects that are missing are key for the interpretation of all results, so they need to be clarified before I could properly assess the quality of the whole paper.

We are sorry to read this and apologize for taking your time. We have now reorganized the method section and have split the results and discussion section to streamline and clarify the manuscript. See L. 116-215 for method description.

- I find it difficult that I cannot see if the actions of adding the substances to the samples itself changed something in the gas concentrations. There are differences in the gas concentrations at t0. This can be attributed to rapid effects of the interaction between the added chemicals and the sample, but it can also be due to sample contamination or degassing during handling. T0 is taken within 24 hours, but it remains unclear if this is after 1 hour for certain samples, and after 23 hours for others. There is also no miliQ or demineralized water control. To have these issues addressed, in a reply to this comment but also in the manuscript, would give me more certainty about the results.

Unfortunately, the experimental design was imperfect, and would have been nice to have some benchmark steps such as pH measurements of the samples right after preservative addition. Now to

overcome these shortcomings, we perform an additional 24h incubation to document the impact of preservatives on pH. See L. 174-187 and results section L. 437-442 and new Fig.2.

Note also that we start the discussion by discussing the validity of using the unfixed samples as representative for "real concentrations". See L. 494-519.

- I would think it benefits the paper if results and discussion are separated into different sections, but I leave it up to the editor to decide on this, as I know there is also personal preference involved on that topic.

This is exactly what we have done to streamline the paper.

- Another key issues that I would like to hear more on is why the t0 concentrations of CO2 of the different treatments is that different. You write some about it, but as I see this as a major factor for many of your interpretations, I would like to have it addressed more in depth and more clearly. Why are there no error bars on the Cu and Hg boxes of CO2?

We now start the discussion by discussing the validity of using the unfixed samples as representative for "real concentrations" and describing and explaining the difference seen at T = 24h. See L. 494-519.

There are no error bars on the Cu and Hg boxes because the points plot all together and that the 25$^{th}$ and 75$^{th}$ are also covered the 100% points, i.e., the 6 replicates. We have now added a description of the box plot in Fig. 1.

- Another important overarching issue is that it seems O2 significantly decreased in the samples with CuCl2. Is this indeed the case? This is of major importance, as it would suggest incomplete inhibition of microbial processes, and would affect all results for the CuCl2 treated samples.

Yes, there is a significant decrease in O2 over time which is likely due to incomplete inhibition of microbial processes. We didn't focus because the acidification of the samples is already a sufficient reason for disapproving its use as preservative. We make a clear point that CuCl2 and HgCl2 should be avoided.

- Specific comments on certain parts of the manuscript are included below.

**Introduction**

- The first paragraph is a bit odd. I think it is most important that the reader knows why it is needed to preserve the samples, not what they are used for in the end. I would put more emphasis on the processes in the bottles that will change the concentration.

Agreed, we have rephrased it. See L. 41-51

- I think it is important to also mention that ZnCl2 and CuCl2 are often used as toxic inhibitors. And give some details on why it seems certain researchers pick which inhibitor (or put that in the discussion). Problems with the disposal of HgCl2 would be good to mention as well, as well as the costs associated with it.

Agreed. However, we didn't put more focus on HgCl2 and gave more space to AgNO3 and CuCl2 in the introduction to re-equilibrate. See L. 75-79 as well as L. 88-89.

- 48 + 49-50. Based on what did you decide to pick these papers as references?

We performed a wide literature search for studies specifically mentioning HgCl2 as preservative for dissolved gas samples in their methods.

- 75 – 81. The tone of this paragraph is also a bit odd. You state quite suddenly that CO2 concentrations will be overestimated by HgCl2, while that is not really clear from the previous paragraphs. And the last line is way too firm for an introduction. The introduction should state the current state of the art, not opinions.

This paragraph has been removed.

- 82 – 92. I would structure this paragraph differently. First, in neutral words (so no effective, overestimation, etc) explain what you investigated. Then state in a few sentence the key findings of your research.

Agreed, we have rephrased the paragraph and hopefully clarified it. See L. 97-111.

**Methods**

- 96. Header 'study area' does not cover the content of the paragraph.

Agreed, we renamed the header to represent the content of the paragraph, see L. 115

- 98. What is carefully collected? Explain in more detail.

The sample procedure is now described in much more detail. See L. 116-131

- 100. Avoid = limit. And not gas loss, but gas exchange with the atmosphere.

Agreed. Corrected.

- 107. That cannot be uniform for all Norwegian lakes, right? Do you mean that is the case for this specific lake, or do you mean this lake was like the lakes tested in that de Wit paper?

Yes, this is now corrected, see L. 130-131

- 112. T=0 could be at any moment during the first 24 hours? Or was it the same for all samples? Were they randomized for the moment of analysis, or were some treatments done first and others later?

This is now clearly described. First time point is T = 24h. See L. 142-147

- 112. Were the same bottles measured at t0, t1 and t2, or were the bottles sacrificed?

This is now clearly described. See L. 139-148

- 113. 'as the water samples were collected'. Which water samples? Is it not the same as the 5L bottles? Unclear.

This sentence has been completely rephrased

- 123. Unclear what 'it' means.

"it" refers to "estimated toxicity". This sentence has been completely rephrased.

- 129. Why were they stored cold? In general, poisoned sampled are not kept cool, as far as I know.

This is how we have worked so far to be on the safe side (see e.g., Clayer et al. 2021), and this also prevents the samples to be subjected to changes in room temperature. Keeping water samples cold and dark is the best way to optimize sample preservation. There is no guarantee that poisoned samples will be 100% inert, keep them cold helps to limit any microbial activity.

- 133. Remove 'unfortunately'.

Done.

- 127. Was the same amount of liquid added to all samples? Was the miliQ flushed to remove the target gasses? What was the volume of the sample bottles?

Yes, the same volume of 240µL was added to all bottles. And no miliQ water was not flushed, but given its small volume, its concentration is unsignificant for the water samples. This is now clarified.

- 141. Sealed with what?

The samples were sealed with gas tight butyl rubber stoppers as for the first experiment. We used the sample bottles and caps for both experiments. This is now clarified L. 153-154.

**Results and discussion**

- Fig. 1. Please split the graphs of O2 and CO2 and CH4 and N2O, it is now not clear that boxes 5-8 are a different gas. 100% saturation at which temperature? That of the lake water, at the 4 degrees of the storage, or the temperature during measurement?

Everything is reported back to *in situ* temperature. This is now clarified in the caption. And the graph has been split. See new Fig. 1

- Table 3. This table is a bit hard to read, while it does contain very interesting information. Can you add some lines or italic or bold text or something, to make it easier for the reader to focus? What is the ice free season number made up of? Please explain in caption. Also write in the caption what diff is (I know it seems obvious but still good to write down).

The style of the journal is to avoid additional line, but we removed the "%" symbol on row 3 and 6, remove the decimal on the row 1 and 2, and arrange the alignments (probably increase the spacing a little) to improve visibility. The ice-free season is the average over the whole experiment duration from April to November. This is now clarified in the caption.

Note also that the "Preservatives" labels were inversed, lower values (1st and 4th rows) are from "unfixed" samples analyzed for DIC while the highest values were those from samples fixed with HgCl2 (2nd and 5th rows).

- Fig. 3. Is the lower panel a useful addition? Or can I already get the same info from the upper panel?

We decided to keep it as it is informative and helps to ensure we use the correct equations and formulations for pH and [H+].

- Fig. 4. Isn't this the exact same info as in table 3? No need to show it twice. I think the graph is much nicer, it brings across your point very clearly.

Table 3 and Fig. 4 both display the monthly mean, yes. In addition, Fig. 4 displays the observations and interpolated daily data. Table 4 also shows the relative difference between fixed and unfixed samples in % as well as the mean for the whole duration of the experiment. Since this is a technical paper, we believe it is useful to keep both for clarity and depending on the reader's affinity for table or figure. However, we agree that we should specifically state that these two objects display the same data. This is now clarified in the captions.

- 357-362. Please address why the t0 concentration in the inhibited samples was so different from the control sample, and why the range in concentrations was much larger. Is it because of sample contamination with air during the addition of the inhibitors? If this is the case, that is not necessarily a bad thing, if you then also explain that that is one of the risks of inhibitor additons to samples.

This is now addressed in the start of the discussion L. 494-519

- 382. Do you suggest that the microbial processes are not inhibited, or that there are abiotic proceses at play? If it's the microbial processes, then how is it possible that these microbes are not inhibited, but the ones using O2 are?

Yes, we suggest that microbial processes are not completely inhibited. Note that certain metabolic pathways can be selectively inhibited and N2O is likely much more sensitive (being at 10's nM levels) to subtle microbial activity than O2 (being at 100's µM levels). Given the none essential character of this information, we decided, after all, to not highlight it. The acidification caused by CuCl2 and HgCl2 is already a sufficient reason to reject its use.

- 386. Please mention whether there were statistically significant changes (between timepoints or between treatments) for N2.

There were no significant changes between timepoints or between treatments for N2. This has been added (L. 435).

- 408. In the CuCl2 treated samples, you have both O2 consumption and CO2 production. Why do you think these are not linked?

Yes, good point. CO2 production in the CuCl2 treated samples is likely partially link to O2 consumption through microbial respiration. However, the CO2 production being much larger than O2 consumption, an additional source of CO2 is needed. This is described L. 525-532

- I have not provided detailed comments on the later sections, as I think it important to know more about the CO2 results first, like I stated in my starting comments.

We hope, we have provided some clarifications. Thank you for your assessment.

**Reviewer 2:**

- General comments
- The technical note describes outcomes from an experiment examining the suitability of three preservatives for the quantification of dissolved gas concentrations, and another

experiment to determine the feasibility of HgCl2 preservation to derive CO2 fluxes from freshwater systems. Despite being toxic, HgCl2 is a commonly used chemical that prevents biological degradation of gas dissolved in water, even though alternatives exist. The study shows that these alternatives are effective and suggests substituting HgCl2 for less toxic preservatives. The results of this study are technically relevant, help reduce and avoid errors in flux estimation and support the implementation of user-friendlier substitutes for the preservation of freshwater samples. I think this study could be very valuable for many researchers in the field and I would like to thank the authors for their nice work. However, the manuscript would benefit from clarifications, and more details especially regarding the Methods are needed before publication.

We are grateful to the reviewer, Thank you so much for this thorough, constructive, and in-depth review. Thank you for pointing out concrete needs for clarifications and improvements. Below we provide response to the main comments, to all numbered comments and technical corrections.

Specific comments:

- Are the studied lakes representative for other lakes or waterbodies? Please clearly outline limitations of this study in terms of impact and application in a broader sense.

Thank you for raising this. Svartkulp is particularly representative of Northern Hemisphere lakes, typically found in granitic bedrock regions in North-East America and Scandinavia. It is a typical low-productivity, heterotrophic, slightly acidic to neutral, moderately humic lake. Similar lakes are found in Southern Norway (de Wit et al., 2023), large parts of Sweden (Valina et al. 2014), and Finland, Atlantic Canada (Houle et al., 2022), Ontario, Québec and North-East USA (Skjelkvåle and de Wit 2011; Weyhenmeyer et al., 2019). Note that even if Svartkulp is among the best buffered lakes in Norway, our findings are also relevant to more acidic, lower ionic strength lakes found in Norway and large parts of Northern Canada.

Lundebyvannet, is also representative of a large group of these Northern lakes, however, it is quite a productive lake with high photosynthetic activity, which is more of a end-member case (e.g., worst-case scenario for Norway, related to CO2 flux overestimation with HgCl2 fixation).

This is now highlighted in the Site description sections L. 131-136 and 214-216, as well as in the discussion (L. 642-646) throughout the manuscript.

- Is it feasible to assume unfixed samples to represent "real" concentrations/fluxes, as control? Could you discuss this further and eventually consider renaming "control" to "unfixed" for the first experiment?

We have reorganized the results and discussion and now start with a discussion on how representatives these unfixed samples are to real conditions. See L. 496-519. For clarity we decided to have a more focused Results section and separate Discussion.

- The Methods section lacks necessary detail. I would suggest to restructure the section to make the experimental setup and respective study lakes clearer. E.g. in the study area section Lake Lundebyvannet should also be introduced, and ideally both lakes should be presented with the same level of detail relevant to the respective experiments. More importantly, I'm missing information on sampling procedures and their feasibility. Since this

is a technical note, I believe the Methods should be sound. I added several comments in this regard below.

Excellent point. We have now described the sampling methods in much more details and present both lakes with a similar level of details. Please see L. 116-158 and 192-213

- I think the results of this study could be put into clear recommendations for future studies, and this could be part of the abstract and expressed more clearly in the discussion.

Thank you for another nice suggestion. We have now added some recommendations to the abstract (L. 32-34) as well as within a designated section in the discussion (L. 648-650 and 664-670)

- The Introduction could benefit from adding some information about the other preservatives studied. The application of HgCl2 is broadly introduced (could be shortened), but the description of the substitutes dealt with in this study falls short. Are there any other studies where CuCl2 or AgNO3 were used to determine dissolved gas concentrations? Are there differences expected between the application of those two?

Agreed, we re-organize and streamlined the introduction. To our knowledge there is no study where CuCl2 or AgNO3 were used to determine dissolved gas concentrations, however, there is one study showing that CuCl2 amendments to soil lowered the pH, we now refer to it. See L. 75-78, as well as L. 89-90.

Please note, the numbers at the beginning of each comment denote the line numbers.

1. 26-27: can you be more specific about time periods (3w, 3m)?

Yes, see now L. 27.

2. 29: are low ionic strength / high DOC lakes representative?

Yes, see L. 30-31

3. 30: are these estimations valid for other lakes?

Yes, see L. 30-31.

4. 31: I think explicitly adding recommendations here would be useful.

Yes, see L. 32-34

5. 59: better in regards to what?

This has been rephrased, see L. 63-65.

6. 67: what is the impact of higher H+ concentrations?

See l. 86-89

7. 70: could you elaborate why this leads to an overestimation of CO2 concentration?

Yes, we will elaborate here. See also l. 86-89

8. 82-92: it would help the reader if it was made clearer here that two different experiments were conducted and two different lakes were sampled for that, for example by 1)... 2)...

Agreed, we rephrased here see L. 97-111

9. 87: This assumes that unfixed samples are the control, or "real results". Is that feasible?

Agreed, we removed the term control and now refer to unfixed or unamended samples, e.g., L. 106-107.

10. 99: Did you collect the water from the surface? Did you use anything other than the bottles to avoid bubbling or degassing? Were samples temperature controlled (or otherwise controlled) between sampling and analysis?

This is now better described, see L. 116-130

11. 100: slowly poured - a bit vague? How could you guarantee no degassing?

This is now better described, see L. 139-158.

12. 103: Are the results you got from the water samples representative for lakes in the region in terms of magnitude? Do the numbers represent means of the sub-samples or was each sub-sample used for determination of one of the parameters?

Yes, see also our response to your 1ˢᵗ comment and see L. 131-136.

13. 105: As far as I understand, the concentration of platinum does not necessarily describe the color characteristics of water?

This has been removed since it is not essential.

14. 106: how did you measure the temperature? is this an important information if the water was transported to the lab? Or did you preserve this temperature during transport?

This is now clarified (L. 128-129), 18.5C was in the lake, we also monitored the temperature in the lab.

15. 111: technically 3 treatments and one control. Which of the scenarios would presumably result in the most "real" concentration?

See our response to 2ⁿᵈ comment above and start of the discussion.

16. 112: why did you choose these time steps? could you elaborate if these times are representative?

These times are now justified, see L. 145-147.

17. 116: Is the preparation of the solutions part of the experiment? do the yielded concentrations have an uncertainty? or would a derivation not have an impact on the outcome?

This is now described L. 159-163.

18. 133: Did the fact that pH was not measured affect your study? Or was that one reason to use the PHREEQC model?

Admittedly, the experimental design was imperfect and pH measurements following fixation should have been included. Now to overcome this, we added a small 24h incubation where pH was measured following same treatment, see L. 174-188 and results L. 438-443 as well as new Fig.2

19. 137: Is the sampling strategy outlined different than the one for the first experiment? How did you achieve sampling water from different depths? It would also be nice if both lakes were described with the same level of detail.

The description of the sampling methods has been revised and should now be clear. See L. 192-213

20. 145: Could you clarify the purpose of DIC analysis in this study?

DIC analyses were performed to obtain an independent estimation of $CO_2$ and DIC concentration, compared to the GC analysis. This is now added at l. 107-110.

21. 147: Add name of TOC analyzer and/or merge with other sections below to avoid repetition. You state that samples were not fixed – why not?

This sentence has been removed.

22. 151: Did you compare the pH data with that measured with the pH-meter as mentioned above, or why measure twice?

We apologize, this is an error. pH was not measured in the laboratory, we only used the pH data from the *in situ* HOBO sensor. This sentence will be corrected.

23. 160: The temperature was recorded during shaking – do you mean the water temperature? What was the purpose?

Yes, the water temperature (sorry for the lack of clarity) was recorded during shaking. This is now clarified with Eq. 1 L. 246-252.

24. 171: Do you mean ambient air was used for calibration? Did you know the concentrations of the ambient air?

Yes, ambient air was used for $O_2$ and $N_2$ calibration. Ambient air is measured regularly and is stable through time.

25. 175 section: I think it would help to directly add formulas in a section in the appendix for better understanding and reproducibility.

Good suggestion, thank you. Eq. 1 was added

26. 187: Could you explain the purpose of DIC analysis here or in earlier sections. Are the $CO_2$ concentrations calculated in addition to the concentrations measured by GC for comparison? I think it may not always be clear where you used measured or calculated $CO_2$ concentrations.

This is introduced earlier l. 107-110.

27. 244: Is it important to mention what files were input and output files? For someone who doesn't know the program, this info seems meaningless.

This is now clarified. See l. 312-313

28. 255: Is this analysis done in retrospect to make up for not measuring sample pH directly after storage (among other things)?

Yes partly. Now we have added a 24h incubation experiment to complement as well. See see L. 174-188 and results L. 438-443 as well as new Fig.2.

29. 320: What temperature did you use to determine the Schmidt number?

This is now clarified. See l. 388

30. 328 f: It is nice to have different temporal resolutions, but what is the purpose of that for this study? Would your measurements not reflect instantaneous fluxes (could maybe be considered as daily fluxes) rather than weekly?

The main idea to show fluxes with these three temporal aggregations is to highlight the magnitude of the mis-estimation of the fluxes when HgCl2 is used as preservatives for the water samples. The error magnitude can be much larger over shorter timescales (see. E.g., Fig. 4).

31. 340: A rather general comment to this study: what is the assumption about the development of gas concentration in between times 0, 3w, 3m? E.g., what would the concentration after 2w or 2m supposedly look like? Did you examine that?

We haven't looked at other time points, this is difficult to predict. We try to only present the data we have and not over interpret. We don't believe there should be any revision related to this comment.

32. 362: Would preservation with AgNO3 then be preferable rather than with HgCl2 due to its toxicity? Can you draw conclusions regarding CH4 from your results?

Yes definitely. These recommendations are now given in the abstract and discussion. L. 496-521

33. 364, Fig. 1: Concentrations of all gases (except CO2) show largest ranges for AgNO3 addition (largest bars) after 3w. Is there an explanation for that? Add to caption: what do the boxplots show, presumably 25th and 75th percentiles and the median?

Unfortunately, no we have no explanation for the relatively larger range for the AgNO3 fixed samples after 3w. We added the description of the boxplots in the caption. See l. 429-430

34. 375 f: Are you arguing that this process is slowed down in freshwater?

Not necessarily slowed down because of freshwaters, there are many parameters playing a potential role here, e.g., temperature, substrate concentration, etc. Rees et al. (2021) performed their incubations at ambient temperatures which is the most likely explanation for the difference seen with our observations. Our samples were stored at 4C for 3 months. This is now clarified. See L. 563-567.

35. 380-381: Is this assumption reflected in your results by the decrease of N2 concentration? Is there also an explanation for the N2O consumption following production?

The interpretation of small changes in the N2 data should be avoided since none of the groups are significantly different from each other. N2O consumption following production has been observed by others, but no specific explanation was proposed.

36. 385: Did you perform a statistical test here too? Is it worthwhile mentioning that the concentrations seem to have the opposite response over time than N2O?

Yes, we did a statistical test for N2, this is now added L. 436. The changes for N2 and N2O do not have the same magnitude, µM for N2, nM for N2O. The expected changes in N2 from the process mentioned in comment #35 is not detectable.

37. 422: The opposite of what you state in the text is shown in Tab. 3. Is there a mistake in the labels?

Yes, there is a mistake in the labels in Table 3, this has now been corrected.

38. 426 f: Wouldn't we expect to see a shift in pH then in Fig. 2 (top panel)? It appears as if the fixed and unfixed samples have the same pH?

We apologize for the confusion. The pH plotted here in the *in situ* pH which is the same for both sample sets. pH was only determined with sensors in situ, this is now clarified in the caption.

39. 435, Tab. 3: What was the reason to show fluxes calculated following Cole and Caraco and not the other wind-based models here?

This was to avoid overloading the table, the values are different but the relative differences between fixed and unfixed samples is bound to the original concentration data. We believe there is no point in showing three models showing the same differences.

40. 466: Why was this cut-off of 20 µM chosen?

This 20 µM cut-off was chosen as the maximum likely error from e.g., pH error of 0.05. This is clarified L. 673-674.

41. 503: Which of those shown in Tab. 3 and Fig. 4 were obtained from DIC analyses?

This is now clarified in the caption of now Fig. 5. See L. 692-693.

42. 507 f: This estimate is only valid for the tested lakes. What would be the implication for other lakes? Is this also valid for sea water samples? Do you have recommendations or a protocol that should be followed? And what about greenhouse gases other than CO2?

Excellent questions, thank you. This is now addressed in section 4.4 from L. 635 as well as section 4.1.

**Technical corrections**

1.      18: what regulations are there? Or do you mean something like "complex handling" instead of regulation?

We refer to the Minemata convention as described in the introduction and conclusion. We clarified L. 18

2.      49: check brackets

Thank you!

3.      66: use abbreviation DOC

Done, thank you.

4.      75: the paragraph could be moved to discussion

Done, thank you.

5.      87: I don't think it's necessary to mention the storage temperature here.

Removed, thank you.

6.      95: determination or rather quantification?

Corrected, thank you.

7.      100: replace "gas loss" with "degassing" throughout.

Corrected, thank you.

8.      106: I think following the journal's guideline you would want to state what NIVA stands for.

Corrected, thank you.

9.      125: silver, not Silver

Corrected, thank you.

10.     132: Add name of gas chromatograph. Maybe it would make sense to merge this section with the Gas chromatography section further below, and move some information to the supplement.

We decided to keep the gas chromatography description below since it also applies to Lundebyvannet samples.

11.     193: I think this description is great, but could be partially merged with sections above and equations moved to a separate section in the supplement.

We preferred to keep everything for clarify, This is a method paper, we decided a thorough description of the methods.

12.     198: pK or K?

These are in fact pK.

13.     205: for completeness state value/equation of K?

The equation can be found in Stumm & Morgan.

14.     206: rather than "given" use "approximated"

Corrected thank you.

15.     208: add altitude "above sea level"

Corrected thank you.

16.     212-214: stick to either air-water or water-atmosphere interface

Corrected thank you.

17.     230: add: in percent

Corrected thank you.

18.     239: without knowing (the power of) this program, I would suggest to move this section or part of it to the supplement, and maybe worth explaining briefly what the program does starting with what PHREEQC stands for. How well does the program perform in predicting variables?

We believe that this section is given with the appropriate level of details and doesn't belong to the SI. It helps the interested reader. In addition, the justification comes early in the description L. 310-312. It is not the place here to describe PHREEQC and what it does.

19.     258: thermodynamically

Corrected thank you.

20.     318: double-check equation numbering

Corrected thank you.

21.     357: The concentration of CH4 [across experiments] ranged…

Corrected thank you.

22.     377: Maybe reword to something like: the prevalence of N2O production in [...] was attributed to favoring more acid conditions.

This sentence has been displaced and rephrased.

23.     411: samples not sampled

This sentence has been displaced and rephrased.

24.     418: indices, not indexes (also change in table)

Corrected, thank you.

25.     432: maybe not necessary to mention Lake Lundebyvannet twice in the caption

Corrected, thank you.

26.     435, Tab. 3: Correct column labels (preservative-addition and None reversed). Remove one "Lake" in caption. Mention what Diff (%) means. No need to include % in each column. Following the journal's guidelines, all figures and tables should be denoted with abbreviated Fig. and Tab. Double-check throughout the manuscript.

Corrected, thank you.

27.     445: as instead of than

Corrected, thank you.

28.     461, Fig. 3: What does i stand for? pH for each sample? I don't think it is needed in the x-axis label then.

Yes, yes it stands for each sample. this has been corrected, thank you.

29.     499: has instead of would have

Corrected, thank you.

30.     503: Fig. S3 shows daily fluxes, not monthly as stated in caption, right?

Yes, this is now corrected, thank you.

31.      505: "in reality" is based on samples without fixation? is that feasible? Did you mean to cite Fig. 2 instead of Fig. 3?

See our response to your second main comment. Yes, we referred to Fig. 2, which is now Fig. 3.

*References:*

Chou W.C., Gong G.C., Yang C.Y. & Chuang K.Y. (2016) A comparison between field and laboratory pH measurements for seawater on the East China Sea shelf. Limnology and Oceanography-Methods, 14, 315-322. https://doi.org/10.1002/lom3.10091

Clayer, F., Thrane, J.-E., Brandt, U., Dörsch, P., & de Wit, H. A. (2021). Boreal Headwater Catchment as Hot Spot of Carbon Processing From Headwater to Fjord. Journal of Geophysical Research: Biogeosciences, 126(12), e2021JG006359. https://doi.org/10.1029/2021JG006359

de Wit, H. A., Garmo, Ø. A., Jackson-Blake, L. A., Clayer, F., Vogt, R. D., Austnes, K., Kaste, Ø., Gundersen, C. B., Guerrerro, J. L., & Hindar, A. (2023). Changing Water Chemistry in One Thousand Norwegian Lakes During Three Decades of Cleaner Air and Climate Change. Global Biogeochemical Cycles, 37(2), e2022GB007509. https://doi.org/10.1029/2022GB007509

Houle, D., Augustin, F., & Couture, S. (2022). Rapid improvement of lake acid–base status in Atlantic Canada following steep decline in precipitation acidity. Canadian Journal of Fisheries and Aquatic Sciences, 79(12), 2126–2137. https://doi.org/10.1139/cjfas-2021-0349

Kokic, J., Wallin, M. B., Chmiel, H. E., Denfeld, B. A., & Sobek, S. (2015). Carbon dioxide evasion from headwater systems strongly contributes to the total export of carbon from a small boreal lake catchment. Journal of Geophysical Research: Biogeosciences, 120(1), 13–28. https://doi.org/10.1002/2014JG002706

Rees, A. P., Brown, I. J., Jayakumar, A., Lessin, G., Somerfield, P. J., & Ward, B. B. (2021). Biological nitrous oxide consumption in oxygenated waters of the high latitude Atlantic Ocean. Communications Earth & Environment, 2(1), Article 1. https://doi.org/10.1038/s43247-021-00104-y

Skjelkvåle, B. L., & de Wit, H. A. (2011). Trends in precipitation chemistry, surface water chemistry and aquatic biota in acidified areas in Europe and North America from 1990 to 2008 (ICP Waters report 106/2011). In 126. Norsk institutt for vannforskning. https://niva.brage.unit.no/niva-xmlui/handle/11250/215591

Sobek, S., Algesten, G., Bergström, A.-K., Jansson, M., & Tranvik, L. J. (2003). The catchment and climate regulation of pCO2 in boreal lakes. Global Change Biology, 9(4), 630–641. https://doi.org/10.1046/j.1365-2486.2003.00619.x

Valiente, N., Eiler, A., Allesson, L., Andersen, T., Clayer, F., Crapart, C., Dörsch, P., Fontaine, L., Heuschele, J., Vogt, R., Wei, J., de Wit, H. A., & Hessen, D. O. (2022). Catchment properties as predictors of greenhouse gas concentrations across a gradient of boreal lakes. 10(880619). https://doi.org/10.3389/fenvs.2022.880619

Valinia, S., Englund, G., Moldan, F., Futter, M. N., Köhler, S. J., Bishop, K., & Fölster, J. (2014). Assessing anthropogenic impact on boreal lakes with historical fish species distribution data and hydrogeochemical modeling. Global Change Biology, 20(9), 2752–2764. https://doi.org/10.1111/gcb.12527

Weyhenmeyer, G. A., Hartmann, J., Hessen, D. O., Kopáček, J., Hejzlar, J., Jacquet, S., Hamilton, S. K., Verburg, P., Leach, T. H., Schmid, M., Flaim, G., Nõges, T., Nõges, P., Wentzky, V. C., Rogora, M., Rusak, J. A., Kosten, S., Paterson, A. M., Teubner, K., … Zechmeister, T. (2019). Widespread diminishing anthropogenic effects on calcium in freshwaters. Scientific Reports, 9(1), Article 1. https://doi.org/10.1038/s41598-019-46838-w

---

## Author Response (AR2)

**Responses to comments on Manuscript Preprint egusphere-2023-1745 (Minor Revisions) by Clayer et al.**

Title: Technical Note: Preventing CO2 overestimation from mercuric or copper (II) chloride preservation of dissolved greenhouse gases in freshwater samples

Referees' comments are underlined for clarity and our responses highlighted in gray. Line numbers refer to line in the revised (track-change) version of the manuscript, if not specified otherwise.

**REFEREE REPORT(S):**

**Referee #1:**

The figure legends are all difficult to understand at the moment. The captions are good. It would help if there were more words and less formulas/technical terms in the legends itself. They can then be explained further in the caption, if there is not enough space in the legend for a complete explanation of what each symbol means exactly.

Thank you for this suggestion. We updated the legend of Fig. 4 (now Fig. 5) which is now believed to be more explicit. Fig 1 doesn't have a legend. The legends in Fig. 2, 3 and 5 are concise and in-line with our terminology in the text. Note that small adjustments were made in Fig. 3 (now Fig. 4) legend changing "pH" for "*in situ* pH".

**Referee #2:**

Thanks to the authors for taking into account the suggested changes, which improved the manuscript and made the experimental setup clearer. I still have some comments with further suggestions. The methods for the two main experiments are described in more detail now, but I am still not understanding parts of them. After some more general concerns, I provide further comments below. The line numbers refer to the document with marked changes.

Thank you for another thorough assessment. Below we provide point by point responses to your suggestions. We have added an overview figure as suggested and have performed minor edits which are believed to significantly improve the clarity of the methods.

**General comments:**

I would like to reiterate the importance of the Methods section in a technical note. I suggest to add a table or graphic showing the two main experimental setups at a glance, together with the main parameters examined, the main instrumentation used, etc. I generally suggest to restructure the Methods section. For example, the GC was used for both experiments, right? However, the DIC analysis only applies to the second experiment if I'm not mistaken. Therefore, I suggest to simply get rid or rename some subsection headings, to clearly distinguish experimental setups. I leave this up to the authors, but currently I have difficulties fully understanding your Methods and experimental setups.

We agree that the Methods section is important. Thank you for a nice suggestion regarding the figure. We have added one (now new Fig. 1) which, we believe, makes the experimental procedures much clearer. The figure highlights where e.g., GC analyses were involved as well as calculations with

PHREEQC. We now start the methods section with one introductory sentence, see L. 116-118 and we also performed minor edits such as line 329 and 342 to improve clarity.

I'm still unsure about the usage of DIC analysis to estimate CO2 concentration. From what I understand this method introduces many uncertainties in estimating CO2 (e.g. Golub et al., 2017, https://doi.org/10.1002/2017JG003794), and I am not really convinced that it is a good way to compare with HgCl2 preserved samples for validation.

Among the various methods available, estimating pCO2 from DIC and pH is the least uncertain method according to Golub et al., 2017. They showed that the relative standard error of pCO2 determination based on DIC and pH was maximum 5.5%. We added this statement in the methods, see L. 210-211.

I also have difficulties with the section about the PHREEQC model and the resulting output. I still think it remains a bit vague what the rationale behind using this model in this study is, how it fits in with the remaining experimental design, and how it contributes to the findings of this study. Should the model outcome be considered essential for this study or is it rather an addition? I would favor moving (at least) the technical description of the model to the appendix and focus on the main components in the Methods section. Can you give references to studies where PHREEQC was used for similar purposes?

As stated in the beginning of the "*Chemical speciation (…)*" in the Methods, PHREEQC was used to estimate effects of preservative on pH, shift in carbonate equilibrium as well as carbonate precipitation (see L.321-323). To support this application, we now refer to Atekwana et al. (2016), Clayer et al. (2016) and Klaus et al. (2023) which have used PHREEQC for similar purposes.

Were results shown in section 3.2 obtained with the PHREEQC model? If so, that should be clarified. Is the pH data shown in Fig. 2 modeled and not directly measured? Are these findings reliable? Does a single comparison as in l. 508 suffice for validation?

These are observations which have been added upon request during revision. See L. 182-195. You are right, several comparison points need to be included, this is now included in section 3.3, see L. 472. Please also note the clarification in the caption of Fig. 3.

I think the additions of explanations in the Discussion helped clarify some aspects of this study and made it much rounder.

Thank you.

**Specific comments:**

l. 57 rewrite to something like: because it proved effective at very low concentrations…

Done, thank you.

l. 63-64 rewrite to: Previous studies showed…

Done.

l. 66 rewrite to: An alternative to using biocides is to collect in-situ water samples, extract the headspace in the field, and analyze the headspace in a laboratory…

Agreed, thank you.

l. 128 and 220: coordinates are missing the degree sign

These coordinates are not expressed as decimals, they do not need a sign since we added the "E" for East and "N" for North.

l. 155 and 205: replace "submitted", e.g. by "lake water from the 25 L bulk sample was subjected to four treatments"

Agreed, thank you.

Tab 1: Could you add the uncertainty to the sample concentrations or can you not quantify it easily?

Yes, we considered the error of the scale (0.01 g) and the error of the flask (0.1 mL) when preparing the solutions.

l. 291: Was the system calibrated before analysis or several times?

The system is calibrated before each run and then standards are run every 5 or 6 samples. This is now added see L. 268.

l. 438: add: samples

Done, thank you.

l. 439: I think it is worth to be precise here. If you state t=24h, this refers to a 24 h incubation time, which was achieved after an equilibration period of the bulk water sample of 24 h, then distributing the bulk water into the bottle volume and leaving it incubate for 24 h, is that correct? Maybe you could briefly clarify that when you first mention this?

As stated L. 159, the filling of the 120mL bottles happened within 3h, the bulk water sample did not equilibrate for more than 1-2h since the lake water was 18.5C and room was at 21C. The total time between sampling and analysis was somewhere around 27 hours, 24h incubation plus 3h of temperature equilibration for the bulk water sample. So given the short equilibration time of the bulk sample, we don't think it is necessary to repeat this information here.

l. 474 and 503: reword to: the whiskers display minimum and maximum

Done, thank you.

l. 498: replace "that" with "the"

Done, thank you.

Fig. 3 lower panel: could the differences we see for CO2 be caused by differences in analysis techniques rather than fixation vs no-fixation? Or how can one be sure this was not the case?

Given that the relative error should be less than 6% (see l. 211), these differences spanning several 100's or even 1000's of µatm cannot be related to analytical biases.

l. 612-613: I do not understand what you mean by "required"

We rephrased this sentence to clarify see l. 532-533.

l. 620 replace "that" with "than"

Done, thank you.

l. 652: please reword to avoid repetition

We rephrased, thank you.

l. 678ff: I am unsure if this is the right location for this explanation or if it belongs in the Results section.

This paragraph fits better in the discussion given the high degree of data analysis and post-processing.

l. 735: what is natural water pH?

We rephrased for clarity, see l. 656-657.

l. 737ff: In terms of impact of this study, I think it would be interesting if you could give the reader an impression of how many studies there are that estimate $CO_2$ concentration or fluxes using $HgCl_2$ (or $CuCl_2$) preservatives in the mentioned regions for freshwater samples? How large would be the overall error of current estimates?

Thank you for a nice suggestion. It is out of the scope of this study to quantify the overall error of current estimates, however, we now point towards a few studies from boreal lakes, but also from tropical aquatic environments where $HgCl_2$ (studies using $CuCl_2$ are rather rare) preservation might be a source of error to raise attention on this problem, see L. 664-669.

l. 767 greenhouse gas concentration

Added, thank you.

**Comments applicable throughout paper:**

Please use consistent names for the chemical addition: you use inhibitors, amendments, biocides, preservative, treatment, etc.

We removed the term "biocides" and only refer to "amendment" for the small volume of preservative solution added to improve clarity.

Replace Figure by Fig. and Table by Tab.

Done, thank you.

I think the overall paper could be shortened and be made more precise to further facilitate readability, but I leave this up to the authors.

We did not find any part that could be shortened and thus decided to leave the manuscript mostly as it is.